# Deciphering functional redundancy in the human microbiome

Liang Tian[1,2,3,4], Xu-Wen Wang [1], Ang-Kun Wu [1,5], Yuhang Fan[1,6], Jonathan Friedman [7], Amber Dahlin[1], Matthew K. Waldor [8,9], George M. Weinstock [10], Scott T. Weiss[1] & Yang-Yu Liu [1]✉

Although the taxonomic composition of the human microbiome varies tremendously across individuals, its gene composition or functional capacity is highly conserved — implying an ecological property known as functional redundancy. Such functional redundancy has been hypothesized to underlie the stability and resilience of the human microbiome, but this hypothesis has never been quantitatively tested. The origin of functional redundancy is still elusive. Here, we investigate the basis for functional redundancy in the human microbiome by analyzing its genomic content network — a bipartite graph that links microbes to the genes in their genomes. We find that this network exhibits several topological features that favor high functional redundancy. Furthermore, we develop a simple genome evolution model to generate genomic content network, finding that moderate selection pressure and high horizontal gene transfer rate are necessary to generate genomic content networks with key topological features that favor high functional redundancy. Finally, we analyze data from two published studies of fecal microbiota transplantation (FMT), finding that high functional redundancy of the recipient's pre-FMT microbiota raises barriers to donor microbiota engraftment. This work elucidates the potential ecological and evolutionary processes that create and maintain functional redundancy in the human microbiome and contribute to its resilience.

---

[1] Channing Division of Network Medicine, Brigham and Women's Hospital and Harvard Medical School, Boston, MA 02115, USA. [2] Department of Physics, Hong Kong Baptist University, Hong Kong SAR, China. [3] Institute of Computational and Theoretical Studies, Hong Kong Baptist University, Hong Kong SAR, China. [4] State Key Laboratory of Environmental and Biological Analysis, Hong Kong Baptist University, Hong Kong SAR, China. [5] Department of Physics and Astronomy, Rutgers University, Piscataway, NJ 08854, USA. [6] Department of Bioengineering, Stanford University, Stanford, CA 94305, USA. [7] Faculty of Agriculture, Food and Environment, Department of Plant Pathology and Microbiology, The Hebrew University of Jerusalem, Jerusalem, Israel. [8] Division of Infectious Diseases, Brigham and Women's Hospital and Harvard Medical School, Boston, MA 02115, USA. [9] Howard Hughes Medical Institute, Boston, MA 02115, USA. [10] The Jackson Laboratory for Genomic Medicine, Farmington, CT 06117, USA. ✉email: yyl@channing.harvard.edu

The human microbiome harbors a plethora of taxa carrying distinct genes and gene families[1], making it functionally diverse. At the same time however, the human microbiome is functionally redundant[2,3], with many phylogenetically unrelated taxa carrying similar genes and performing similar functions[4–7]. For example, dietary carbohydrates can be metabolized by either *Prevotella* (from the phylum Bacteroidetes) or *Ruminococcus* (from the phylum Firmicutes)[8]. Short-chain fatty acids can be produced by multiple common genera including *Phascolarctobacterium*, *Roseburia*, *Bacteroides*, *Clostridium*, *Ruminococcus*, etc[9]. Bile acids can be modified by bacteria belonging to Lachnospiraceae, Clostridiaceae, Erysipelotrichaceae, and Ruminococcaceae[10]. Interleukin secretion can be promoted by *Sutterella*, *Akkermansia*, *Bifidobacterium*, *Roseburia*, and *Faecalibacterium prausnitzii*[11,12]. Moreover, several metagenomic studies have reported that the carriage of microbial taxa varies tremendously within healthy populations, whereas microbiome gene compositions or functional profiles remain remarkably conserved across individuals[1,13–16]. Despite the functional variations and microbial gene diversity that have been uncovered through refined computational metagenomic processing[17] and meta-analysis[18], the highly conserved functional profiles across individuals imply significant functional redundancy (FR) in the human microbiome.

It has been suggested that this significant FR underlies the stability and resilience of the human microbiome in response to perturbations[2,19], but there is little evidence to substantiate this idea. The origin of the FR observed in the human microbiome is still not well understood. A paradox has been raised recently, based on the fact that selection pressures could operate at different levels in the human-microbial hierarchy[20], which potentially could drive the FR of the human microbiome in opposite directions. From the host perspective, although strong FR does not necessarily imply that the host is regulating the diversity of microbiota to promote FR[21], host-driven or "top-down" selection would result in a community composed of widely divergent microbial lineages whose genomes contain functionally similar suites of genes, leading to high FR within the community. From the microbial perspective, species with similar genomes (functional capacities) will tend to occupy the same ecological niche and hence compete with each other. Such competitions between members of the microbiota would exert "bottom-up" selection pressure that results in specialized genomes with functionally distinct suites of genes, leading to high functional diversity (FD) and low FR within the community. This apparent paradox is oversimplified, as it doesn't take into account the spatial structure and heterogeneous environments inhabited by the human microbiome. Nevertheless, low FR will tend to arise from widely divergent microbial lineages with functionally distinct suites of genes inhabiting the diverse niches within host body sites. On the other hand, high FR will arise from the presence of a core or common set of genes, i.e., housekeeping genes, required for diverse microbes to perform basic cellular functions and/or survive in the host body site they inhabit.

Here, we investigate whether there is any organizing principle or assembly rule of the human microbiome that explains the observed high level of FR. In particular, we constructed the genomic content network (GCN) of the human microbiome, which is a bipartite graph connecting microbes to the genes in their genomes. The GCN provides a full description of the functional overlap of different microbes in microbial communities, which enables us to quantify the within-sample FR for any given human microbiome sample for the first time. Then we applied tools from network science[22] to study the topological features of the GCN that determine the FR of human microbiome samples. Furthermore, we developed a simple genome evolution model that can reproduce all the key topological features of the GCN. Using this model, we identified key evolutionary and ecological factors that account for the topological features of the GCN, and hence revealed the origin of FR in the human microbiome.

## Results

**Definition of within-sample FR.** Consider a pool of $N$ taxa, which contains a collection of $M$ genes. The microbial composition or taxonomic profile $\boldsymbol{p}^{(\nu)} = \left[p_1^{(\nu)}, \cdots, p_N^{(\nu)}\right]$ of a local community $\nu$ (i.e., a microbiome sample from a particular body site of a human subject $\nu$) can be directly related to its gene composition or functional profile $\boldsymbol{f}^{(\nu)} = \left[f_1^{(\nu)}, \cdots, f_M^{(\nu)}\right]$ through the GCN of the metacommunity (Fig. 1a–c). Here, we define the GCN as a weighted bipartite graph connecting these taxa to their genes. The GCN can be represented by an $N \times M$ incidence matrix $\boldsymbol{G} = (G_{ia})$, where a non-negative integer $G_{ia}$ indicates the copy number of gene $a$ in the genome of taxon-$i$ (Fig. 1b). The functional profile is given by $\boldsymbol{f}^{(\nu)} = c\boldsymbol{p}^{(\nu)}\boldsymbol{G}$, where $c = \left[\sum_{a=1}^{M}\sum_{i=1}^{N} p_i^{(\nu)} G_{ia}\right]^{-1}$ is a normalization constant (see Methods).

A key advantage of GCN is that it enables us to calculate the FR for each local community, i.e., the within-sample or alpha FR (hereafter, denoted as $FR_\alpha$). In the ecological literature, the FR of a local community is often interpreted as the part of its alpha taxonomic diversity ($TD_\alpha$) that cannot be explained by its alpha functional diversity ($FD_\alpha$)[23–25]; i.e.,

$$FR_\alpha \equiv TD_\alpha - FD_\alpha. \tag{1}$$

Typically, $TD_\alpha$ is chosen to be the Gini-Simpson index:

$$GSI \equiv 1 - \sum_{i=1}^{N} p_i^2 = \sum_{i=1}^{N}\sum_{j \neq i}^{N} p_i p_j, \tag{2}$$

representing the probability that two randomly chosen members of the local community (with replacement) belong to two different taxa; and $FD_\alpha$ is chosen to be the Rao's quadratic entropy

$$Q \equiv \sum_{i=1}^{N}\sum_{j \neq i}^{N} d_{ij} p_i p_j, \tag{3}$$

a classical alpha diversity measure that characterizes the mean functional distance between any two randomly chosen members in the local community[23,24]. Here, $d_{ij} = d_{ji} \in [0,1]$ denotes the functional distance between taxon-$i$ and taxon-$j$, which can be calculated as the weighted Jaccard distance between the genomes of the two taxa (see Methods and Supplementary Fig. 1 for other definitions of $d_{ij}$). By definition, $d_{ii} = 0$ for $i = 1,\ldots, N$. Note that with $TD_\alpha = GSI$ and $FD_\alpha = Q$, we have

$$FR_\alpha = \sum_{i=1}^{N}\sum_{j \neq i}^{N}(1 - d_{ij}) p_i p_j, \tag{4}$$

naturally representing the functional similarity (or overlap) of two randomly chosen members in the local community. From Eq. (4), we can see clearly that $FR_\alpha$ of any microbiome sample is jointly determined by two factors: (1) the functional distances $d_{ij}$'s among taxa present in the sample, which are predetermined by the structure of the GCN; and (2) the microbial composition or taxonomic profile $\boldsymbol{p} = [p_1,\ldots, p_N]$ of this microbiome sample. Of course, we can also use other definitions for $TD_\alpha$ and $FD_\alpha$, then the expression of $FR_\alpha$ will be different. In particular, we can consider a parametric class of taxonomic (or functional) diversity

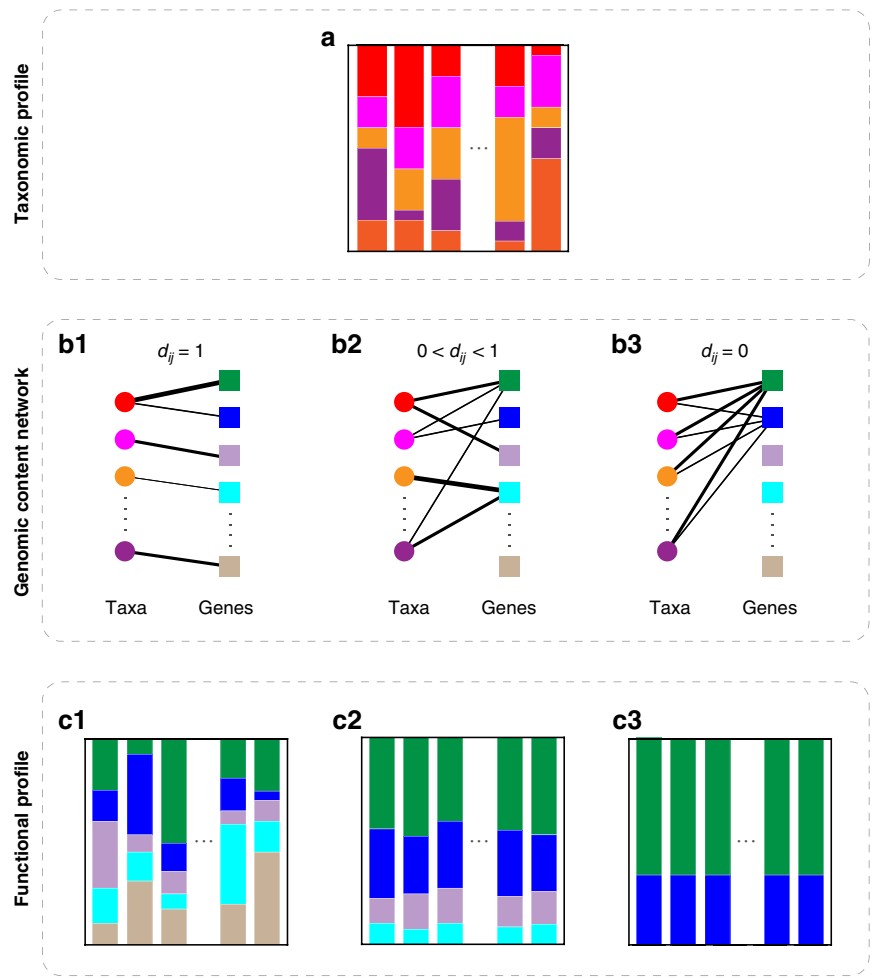

**Fig. 1 Structure of the genomic content network is crucial for determining the functional redundancy and functional diversity of microbial communities.** Here we use hypothetical examples to demonstrate this point. **a** The taxonomic profiles vary drastically across many local communities (i.e., microbiome samples from different individuals). **b** Genomic content networks are bipartite graphs that connect taxa to the genes in their genomes. The left-hand side nodes (circles) represent different taxa and the right-hand side nodes (squares) represent different genes. The edge weight represents the gene copy number. **b-1** Each taxon has a unique genome. **b-2** Different taxa share a few common genes, some taxa are specialized to have some unique genes. **b-3** All taxa share exactly the same genome. **c** For each microbiome sample, its functional profile can be calculated from its taxonomic profile in **a** and the genomic content network in **b**. **c-1** The functional profiles vary drastically across different microbiome samples. For each sample, the functional diversity is maximized while the functional redundancy is minimized. **c-2** The functional profiles are highly conserved across different samples. The within-sample functional diversity and functional redundancy are comparable. **c-3** The functional profiles are exactly the same across all different microbiome samples. For each sample, the functional diversity is minimized while the functional redundancy is maximized.

measures based on Hill numbers[26,27]. Even in this case $FR_\alpha$ of any microbiome sample is still jointly determined by the structure of the GCN and the microbial composition of the sample. Also, we have confirmed that this does not affect our main results presented below (see Supplementary Sec. 1 and Supplementary Fig. 2 for details).

The $FR_\alpha$ of each local community (or microbiome sample) is closely related to the system-level FR observed over a collection of samples. Consider two extreme cases: (i) each taxon is completely specialized and has its own unique genome (Fig. 1b1), hence $d_{ij} = 1$ for any $i \neq j$. In this case, for each sample we have $FD_\alpha = TD_\alpha$ and $FR_\alpha = 0$. The functional profiles vary drastically across samples (Fig. 1c1). (ii) All taxa share exactly the same genome (Fig. 1b3), rendering $d_{ij} = 0$ for all $i$ and $j$. In this case, for each sample we have $FD_\alpha = 0$ and $FR_\alpha = TD_\alpha$. The function profiles are exactly the same for all samples (Fig. 1c3). These two extreme scenarios are of course unrealistic. In a more realistic inter-mediate scenario, the GCN has certain topological features such that different taxa share a few common functions, but some taxa

are specialized to perform some unique functions (Fig. 1b2). In this case, the $FD_\alpha$ and $FR_\alpha$ of each sample can both be high. Moreover, the functional profiles can be highly conserved across samples (Fig. 1c2).

Note that the genotype–phenotype mapping is relatively simple for prokaryotes, which enables us to relate their gene content and functional capacity. For higher organisms, their gene content and functional capacity are not simply related, which means that the GCN framework presented here cannot be simply applied to study the FR of communities of higher organisms.

**A reference GCN.** Although the taxonomic profiles of human microbiome samples are highly personalized, we can construct a reference GCN based on the pool of human-associated microbes to quantitatively study the GCN underlying the human micro-biome. Here we constructed a reference GCN using the Integrated Microbial Genomes & Microbiomes (IMG/M) database[28], focusing on the Human Microbiome Project (HMP) generated metagenome datasets[29]. The IMG/M-HMP database used here

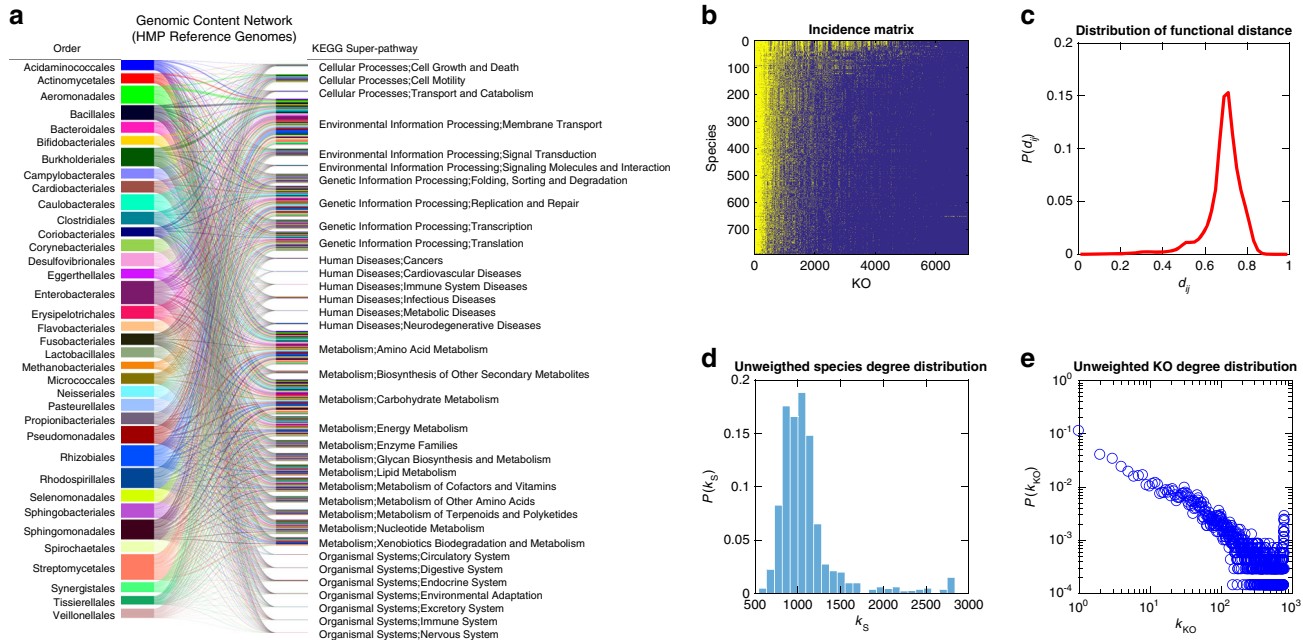

**Fig. 2 The genomic content network (GCN) constructed from the Integrated Microbial Genomes and Microbiome (IMG/M) database has nested structure and heterogeneous gene degree distribution.** We use IMG/M-HMP, an IMG/M data mart that focuses on the Human Microbiome Project (HMP) generated metagenome data sets[29] to construct the GCN. **a** For visualization purpose, we depict this reference GCN at the order level for taxon nodes and at the KEGG super-pathway level for function nodes. The bar height of each order corresponds to the average genome size of those species belonging to that order. The thickness of a link connecting an order and a KEGG super-pathway is proportional to the number of KOs that belong to that super-pathway, as well as the genomes of species in that order. The majority of the super-pathways shown here are related to the metabolic, environmental, and genetic processes performed by microbes. However, for a small number of taxa, as some of their genes have mammalian and/or human disease orthologs, we also identified several super-pathways involved in human diseases and higher-order organizational systems. See Supplementary Sec. 2.1 for the details of constructing this reference GCN. **b** The incidence matrix of this reference GCN is shown at the species-KO level, where the presence (or absence) of a link between a species and a KO is colored in yellow (or blue), respectively. We organized this matrix using the Nestedness Temperature Calculator to emphasize its nested structure[31]. The nestedness value (~0.34712) of this network is calculated based on the classical NODF measure[32] (see Methods for details). **c** The probability distribution of functional distances ($d_{ij}$) among different species. The bin size is 0.02. **d** The unweighted species degree distribution. Here, the unweighted degree of a species is the number of distinct KOs in its genome. **e** The unweighted KO-degree distribution. Here, the unweighted degree of a KO is the number of species whose genomes contain this KO.

includes in total 1555 strains and 7210 KEGG Orthologs (KOs) (see Supplementary Sec. 2.1.1 for details). Here, each KO is a group of genes representing functional orthologs in molecular networks[30]. In order to reduce the culturing and sequencing bias for certain species (e.g., *Escherichia coli*), we randomly chose a representative strain (genome) for each species, which results in a reference GCN of 796 species and 7105 KOs. This reference GCN is depicted in Fig. 2a as a bipartite graph, where for visualization purposes each taxon node represents an order and each function node represents a KEGG super-pathway.

In order to characterize the structure of this reference GCN, we systematically analyzed its network properties at the species-KO level. We first visualized its incidence matrix (Fig. 2b), where the presence (or absence) of a link connecting a species and a KO is colored in yellow (or blue), respectively. We noticed that this matrix displays a highly nested structure[31–33], i.e., the KOs of those species in the lower rows (with smaller genome size) tend to be subsets of KOs for those species in the higher rows (with larger genome size). The nestedness of the GCN can be quantified using the classical NODF measure[32], and turns out to be much higher than expected by chance. (See Methods and Supplementary Figs. 3, 5, 8 for details.) We then calculated the functional distances among different species, finding a unimodal distribution with the peak centered around 0.7 (Fig. 2c). Finally, the unweighted degree distributions of taxon nodes (species) and function nodes (KOs) were calculated. Here, the unweighted degree of a species is just the number of distinct KOs in its

genome, and the unweighted degree of a KO is the number of species whose genomes contain this KO. We found that the unweighted degrees of species follow a Poisson-like distribution (Fig. 2d), implying that in general, species contain very similar numbers of distinct KOs. By contrast, the unweighted degree distribution of KOs is highly heterogeneous and displays a fat tail (Fig. 2e), indicating that most KOs are specialized and only exist in the genomes of very few species, and a few housekeeping KOs appear in almost every species' genome to maintain basic cellular functions. (Note that these housekeeping KOs also appear as the leftmost yellow columns in the incidence matrix shown in Fig. 2b.) This is consistent with the characteristic asymmetrical U-shape observed in the gene frequency distributions of prokaryotic pangenomes[34,35]. Analyses of the reference GCN constructed by using other genome annotation, e.g., Clusters of Orthologous Groups of proteins (COGs)[36], or constructed from a different database (MBGD: Microbial Genome Database for Comparative Analysis)[37] revealed very similar network properties (Supplementary Fig. 3) and did not affect our main results presented below (Supplementary Fig. 4).

The highly nested structure of the reference GCN is intriguing. This structure cannot be simply accounted for by housekeeping genes or the U-shape gene degree distribution. First, as shown in Fig. 2b, the incidence matrix of the GCN still displays a highly nested structure even in the absence of housekeeping genes (the leftmost yellow columns). Second, if we randomize the GCN but preserve the gene degree distribution, the randomized GCNs have

much lower nestedness than that of the real GCN (Supplementary Figs. 3, 5, 8). Third, we adopted tools from statistical physics to calculate the expected nestedness value and its standard deviation for an ensemble of randomized GCNs in which the expected species and gene degree distributions match those of the real GCN[38]. We found that the expected nestedness of randomized GCNs is significantly lower than that of the real GCN (one sample $z$ test yields $p_{\text{value}} = 6.2853 \times 10^{-5}$, see Methods for details).

**Within-sample FR calculation based on reference genomes.** Using shotgun metagenomic sequencing data from two large-scale microbiome studies, the HMP[13,39,40] and the MetaHIT (Metagenomics of the Human Intestinal Tract)[1,41], we calculated the FR of human microbiome samples collected from different body sites. First, we constructed body site-specific GCNs using the IMG/M-HMP database (see Supplementary Sec. 2.1.2 for details). Note that the body site-specific GCNs display similar network properties as the global reference GCN constructed from the IMG/M-HMP database (Supplementary Fig. 5). To remove the potential impact of body site-dependent $TD_\alpha$ on the calculated $FR_\alpha$, we computed the normalized $FR_\alpha$ (i.e., $nFR_\alpha \equiv FR_\alpha/TD_\alpha$) for these samples. Interestingly, we found that in both HMP and MetaHIT studies and for most body sites $nFR_\alpha \sim 0.4$ (Fig. 3a, b, black boxes), suggesting that $FR_\alpha$ and $FD_\alpha$ are generally comparable for human microbiome samples. We also confirmed that the results are not sensitive to the integrity of the KEGG database, since $nFR_\alpha$ is stable if we randomly remove KOs from the GCN (Supplementary Fig. 6). Moreover, additional analyses demonstrated that although housekeeping KOs contribute to higher FR values, they are not the primary explanation for FR (Supplementary Fig. 7).

**Disentangle impacts of GCN and microbial composition on FR.** As mentioned above, $FR_\alpha$ of any microbiome sample is jointly determined by two factors: (1) the functional distances $d_{ij}$'s among taxa present in the sample that are predetermined by the structure of the GCN; and (2) the microbial composition $\boldsymbol{p} = [p_1,...,p_N]$ of this sample. Yet, this does not mean mathematically one can separate the $FR_\alpha$ of any microbiome sample into two independent and additive terms: one is purely contributed by GCN, and the other is purely contributed by the microbial composition. Indeed, as shown in Eq. (4) (or Eqs. [S20–S21]), there is always a term in $FR_\alpha$ that involves the multiplication of $d_{ij}$ and $p_i p_j$ (or their respective functions). This term cannot be separated into two independent and additive expressions of $d_{ij}$ and $p_i p_j$, respectively. To study which of the two factors plays a more important role in determining the $FR_\alpha$ of microbiome samples, we have to "disentangle" the impacts of the two factors on $FR_\alpha$ in a more sophisticated way. To achieve that, in the following two subsections, we introduced two different types of null models: null-GCN models and null-composition models.

**Impact of GCN structure on within-sample FR.** To study the impact of GCN on the within-sample FR of a microbiome sample, we can fix its microbial composition and then randomize the GCN. To identify key topological features of the GCN that determine $nFR_\alpha$, we adopted tools from network science. In particular, we randomized the body site-specific GCNs using four different randomization schemes, yielding four different null-GCN models (see Supplementary Sec. 3.1 for details). Then, we recalculated $nFR_\alpha$ for each sample (Fig. 3a, b, colored boxes), finding that for all the body sites examined all the four different null models yield lower $nFR_\alpha$ than those calculated from real

body site-specific GCNs (Fig. 3a, b, black boxes). Analyzing the network properties of those null models (Supplementary Fig. 8), we found that those randomized GCNs all display lower nestedness and higher $d_{ij}$ than those of the real GCNs. Thus, the highly nested structure and low $d_{ij}$ of the real GCNs contribute to the high $nFR_\alpha$ values observed in the microbiome samples. Moreover, for the first two null models (Null-GCN-1 and Null-GCN-2, where both the highly nested structure and high gene degree heterogeneity of the real GCN are destroyed), $nFR_\alpha$ is much lower than those of the other two null models (Null-GCN-3 and Null-GCN-4, where the highly nested structure is destroyed, but the high gene degree heterogeneity is kept). This suggests that the high gene degree heterogeneity also contributes to the high $nFR_\alpha$ values of those microbiome samples. Hence, the GCN exhibits at least three different topological features (highly nested structure, low $d_{ij}$, and heterogeneous gene degree distribution) that jointly contribute to the high $nFR_\alpha$ value of microbiome samples. We emphasize that these findings do not depend on the detailed definitions of $d_{ij}$, $FR_\alpha$, $FD_\alpha$, or the functional annotation of genomes (Supplementary Figs. 1, 2, 4).

**Impact of microbial composition on within-sample FR.** To study the impact of microbial composition on the within-sample FR of a microbiome sample, we can fix the GCN, and then randomize the microbial composition. In particular, to test if the microbe assemblages or their abundances play an important role in determining $nFR_\alpha$, we randomized the taxonomic profiles using three different randomization schemes, yielding three different null-composition models (see Supplementary Sec. 3.2 for details). Then, we recalculated $nFR_\alpha$ for each sample (Fig. 3c, d, colored boxes). We found that for each microbiome sample if we preserve the abundance profile but randomly replace the species by those present in the species pool (i.e., in the corresponding body site-specific GCN), the resulting null-composition model (Null-compostion-1) always yields much lower $nFR_\alpha$ than that of the original sample. This suggests that the species present in each microbiome sample are not assembled at random, but follow certain functional assembly rules[42]. Interestingly, if we randomize the microbial compositions through random permutation of non-zero abundance for each sample across different species (Null-composition-2) or for each species across different samples (Null-composition-3), those two null models did not always yield much lower $nFR_\alpha$ than that of the original sample. Again, these observations do not rely on the detailed definitions of $d_{ij}$, $FR_\alpha$, $FD_\alpha$, or the functional annotation of genomes (Supplementary Figs. 1, 2, 4). These observations suggest that the assemblage of microbes plays a more important role than their abundances in determining the high FR of the human microbiome. We hypothesize that the specific environment (e.g., the host nutrient and immune state) from which particular microbiome samples were obtained will tend to select for sets of functions among most or all inhabitants, at any abundance. This could partially explain why assemblage or membership matters more than abundances in determining FR.

Note that, for Null-composition-2 and Null-compoisition-3, the sample-specific GCN is fixed, whereas for Null-compoisition-1, the sample-specific GCN is actually different from that of the real microbiome sample (because species in the sample are randomly replaced by species from the species pool). But we argue that the key structure features (e.g., highly nested structure, low $\langle d_{ij} \rangle$, and heterogeneous gene degree distribution) are still preserved, even after the species replacement. In other words, Null-compoisition-1 still preserves the key structure features of the sample-specific GCN, which also reflect the features of the reference GCN.

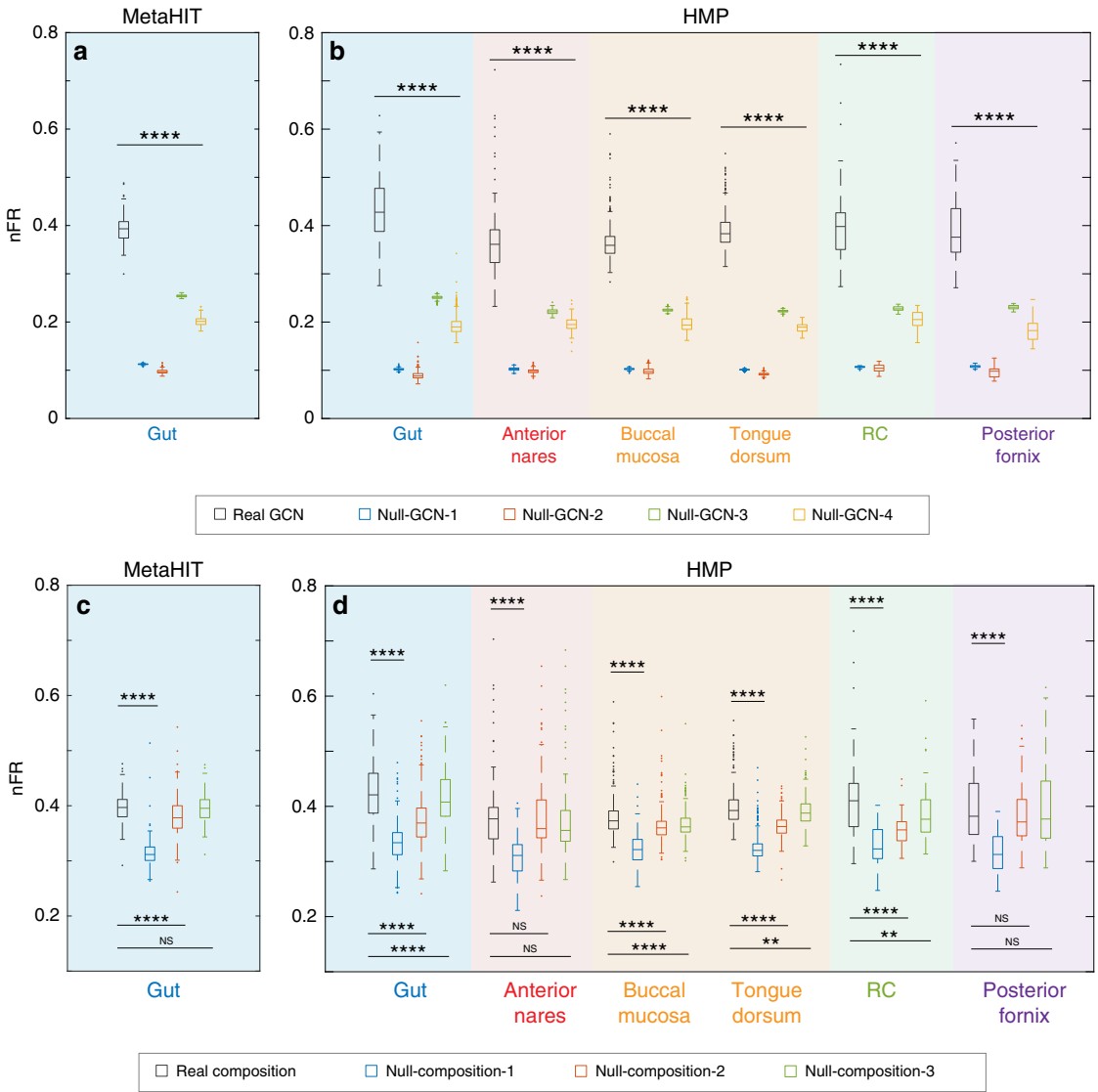

**Fig. 3 Topological features of the genomic content network and the assemblage pattern in the human-associated microbial communities contribute to the high functional redundancy observed in the human microbiome.** Shotgun metagenomic sequencing data from HMP[13,39,40] (for six different body sites: gut, $n = 549$ samples; anterior nares $n = 87$ samples; buccal mucosa $n = 368$ samples; tongue dorsum, $n = 418$ samples; retroauricular crease, RC, $n = 36$ samples; posterior fornix $n = 52$ samples) and MetaHIT[1,41] (for gut, $n = 177$ samples) were analyzed. See Methods for detailed descriptions of the two metagenomic data sets. **a, b** The box plots of the normalized function redundancy ($nFR_\alpha \equiv FR_\alpha/TD_\alpha$) were calculated from the real GCN (black box), as well as the randomized GCNs (colored boxes) using four different randomization schemes: Complete randomization (Null-GCN-1); Species degree preserving randomization (Null-GCN-2); KO-degree preserving randomization (Null-GCN-3); Species- and KO-degree preserving randomization (Null-GCN-4). See Supplementary Sec. 3.1 for details of these randomization schemes. Here the (weighted) degree of a KO is the sum of copy numbers of this KO in those genomes that contain it, and the (weighted) degree of a species is the sum of copy numbers of those KOs in this species' genome. **c, d** The box plots of normalized function redundancy were calculated from the real microbial compositions (black box), as well as the randomized microbial compositions (colored boxes) using three different randomization schemes: Randomized microbial assemblage generated by randomly choosing the same number of species from the species pool but keeping the species abundance profile unchanged (Null-composition-1); randomized microbial abundance profiles through random permutation of non-zero abundance for each sample across different species (Null-composition-2); randomized microbial abundance profiles through random permutation of non-zero abundance for each species across different samples (Null-composition-3). See Supplementary Sec. 3.2 for details of the randomization schemes. Boxes indicate the interquartile range between the first and third quartiles with the central mark inside each box indicating the median. Whiskers extend to the lowest and highest values within 1.5 times the interquartile range. Statistical analysis was performed using the two-sided Wilcoxon signed rank test. Significance levels: FDR-corrected $p$ value $< 0.05$ (*), $<0.01$(**), $<0.001$(***), $<0.0001$(****); $>0.05$ (NS, non-significant). See Source data for the exact FDR-corrected $p$ values.

**Within-sample FR calculation based on de novo taxonomic profiling.** All the results calculated from shotgun metagenomic sequencing data presented above are based on taxonomic profiling using existing reference genomes. To test if our findings could be derived independent of reference genomes, we adopted a de novo method to perform taxonomic profiling of shotgun metagenomic sequencing data without using any reference genomes[41]. This de novo taxonomic profiling method is based on the binning of co-abundant genes across a series of metagenomic samples. We applied this method to the human gut microbiome samples from MetaHIT to construct a GCN (see Supplementary Sec. 2.2 for details). Notably, we found that this GCN displays

very similar network properties as the GCN constructed using reference genomes, i.e., high nestedness, a unimodal functional distance distribution with a clear peak centered ~0.7, Poisson-like species degree distribution, and a fat-tailed gene degree distribution (Supplementary Fig. 9). Using the taxonomic profiles and the constructed GCN obtained from this method, we further calculated the normalized FR of real microbiome samples and compared these values to those calculated from randomized GCNs or randomized microbial compositions (Supplementary Fig. 10). We found that all the key findings presented in Fig. 3 can be reproduced, implying that our results do not depend on the existing reference genomes.

**A simple genome evolution model**. To gain more biological insight into the bases of the topological features of the real GCN, and thus deepen understanding of the origin of FR in the human microbiome, we developed a simple genome evolution model. In this model, we explicitly considered selection pressure and the processes of gene gain and loss, and horizontal gene transfer (HGT) (Fig. 4a). (See Supplementary Sec. 4 and Supplementary Figs. 11–13 for details.) To offer a minimal model, we assumed selection pressure simply favors changes in larger genomes. We found that with reasonable model parameters all the key topological features of the real GCN can be reproduced by our simple model (Fig. 4b–e). Moreover, we found that a high HGT rate is necessary to generate a GCN with a highly nested structure (Fig. 4f) and a very heterogeneous gene degree distribution as observed in the real GCN (Fig. 4g), which are crucial features to maintain high FR in the human microbiome. As shown in Fig. 4f, the nestedness (measured by NODF) of the GCN generated by our model displays a phase-transition like behavior: when the HGT rate is above certain threshold value, NODF deviates from zero and increases gradually. Similarly, as shown in Fig. 4g, the Kullback–Leibler (KL) divergence between the normalized gene degree distribution of real GCN and that of a simulated GCN also displays a phase-transition like behavior. When the HGT rate is above certain threshold value, the KL divergence drops and becomes very close to zero, implying that the gene degree distribution of the generated GCN is very similar to that of the real GCN. These results highlight the importance of HGT in determining the high FR of the human microbiome. In SI Supplementary Fig. 13, we further demonstrated that both the incidence matrix of GCN and the functional distance distribution will be quite different from that observed in the real GCN, if the selection pressure is zero or too large. This implies that moderate selection pressure is needed to reproduce key topological features of the GCN, and consequently favors high FR.

**Within-sample FR as a resilience indicator**. It has been suggested that the strong FR found in the human microbiome is basis for the stability and resilience of its response to perturbations[2,19]. This hypothesis is largely based on the following consideration. An ecosystem with higher level of FR will be more resistant to the addition of new species, because any newly added species will very likely be functionally similar to certain existing species. Owing to the Competitive Exclusion Principle[43], those newly added species will fail in the competition with their functionally similar species, rendering poor engraftment. Although theoretically reasonable, there is no overwhelming evidence yet to directly validate this hypothesis using real data.

The GCN-based framework allows us to quantify within-sample FR and hence quantitatively test this intriguing hypothesis. To demonstrate this promise, we analyzed microbiome data from two fecal microbiota transplantation (FMT) studies[44,45] to check if the FR level of the recipient's pre-FMT microbiota is

related to the donor microbiota engraftment. In both studies, to quantify the extent of donor microbiota colonization after FMT, shotgun metagenomic sequencing was performed to quantify and characterize the extent of changes to the structure of the gut microbiome after FMT[44,45]. For each individual in the two FMT studies, we plotted the fraction of donor-specific strains (denoted as $f_{ds}$) as a function of (1) the time post-FMT (denoted as $t_{post}$); and (2) the TD (FD or FR) of the pre-FMT gut microbiota, denoted as $TD_{pre}$ ($FD_{pre}$ or $FR_{pre}$, respectively) (see Fig. 5). Multiple linear regression with $F$ test revealed significant negative association between $f_{ds}$ and $FR_{pre}$ (or $TD_{pre}$, but not $FD_{pre}$) in both studies. Moreover, the negative association between $f_{ds}$ and $FR_{pre}$ is much stronger than that between $f_{ds}$ and $TD_{pre}$ (or $FD_{pre}$). These results suggest that high FR of the recipient's pre-FMT microbiota raises barriers to donor microbiota engraftment, presumably reducing FMT efficacy; whereas low FR is expected to reduce the resilience of the pre-FMT gut microbiota against external perturbation, potentially facilitating the efficacy of FMT in restoring a healthy gut microbiota. Despite some limitations (e.g., the small sample sizes and the potential donor-recipient compatibility issue), this result is consistent with our hypothesis. Moreover, it suggests that the FR of the human microbiome may serve as a resilience indicator in response to perturbations such as FMT. A more rigorous investigation of FR as a resilience indicator of the human microbiome warrants more dedicated clinical studies, which are beyond the scope of this paper.

## Discussion

In sum, we developed a GCN-based framework to quantify the FR of the human microbiome and revealed the origin of FR using a genome evolution model. The GCN framework enabled us to directly validate if a strong FR underlies the stability and resilience of the human microbiome in response to perturbations[2,19], such as FMT. This could potentially inform other microbiome-based therapies such as probiotic administration, if FR can indeed serve as a residence indicator of the human microbiome in response to general perturbations. FR has been found in many other microbial systems as well, e.g., in plant[46,47], ocean[48], and soil[49,50] microbiomes. Our general, quantitative measure of FR can also be directly applied to those microbial systems and hence facilitate a direct test of the hypothesis that there are systematic differences in FR between free-living and host-associated microbial communities[51]. More broadly, we anticipate that the GCN framework will yield new insights into the relationships between biodiversity and ecosystem function for diverse microbial communities.

In an ecological network, the importance of a species can be quantified by measuring the centrality[52] of its position in the network, where nodes represent different species and edges represent direct ecological interactions between different species (e.g., parasitism, commensalism, mutualism, amensalism, or competition)[53,54]. We emphasize that the GCN defined here is fundamentally different from the ecological networks in literature. In the GCN, nodes represent species and genes, and links represent the presence (and copy number) of a gene in the genome of a particular species. It is very challenging, if not impossible, to infer inter-species interactions based on the GCN because there is clear relationship between the genome similarity of different species and their ecological interactions. Similarly, it might be very challenging to infer species abundance correlation[55] or co-occurrence[56,57] simply based on the GCN.

In the current work, our primary goal was to establish the GCN framework, validate the computation pipeline of within-sample FR calculation, and explain the high FR observed in the human microbiome, using cross-sectional shotgun metagenomic

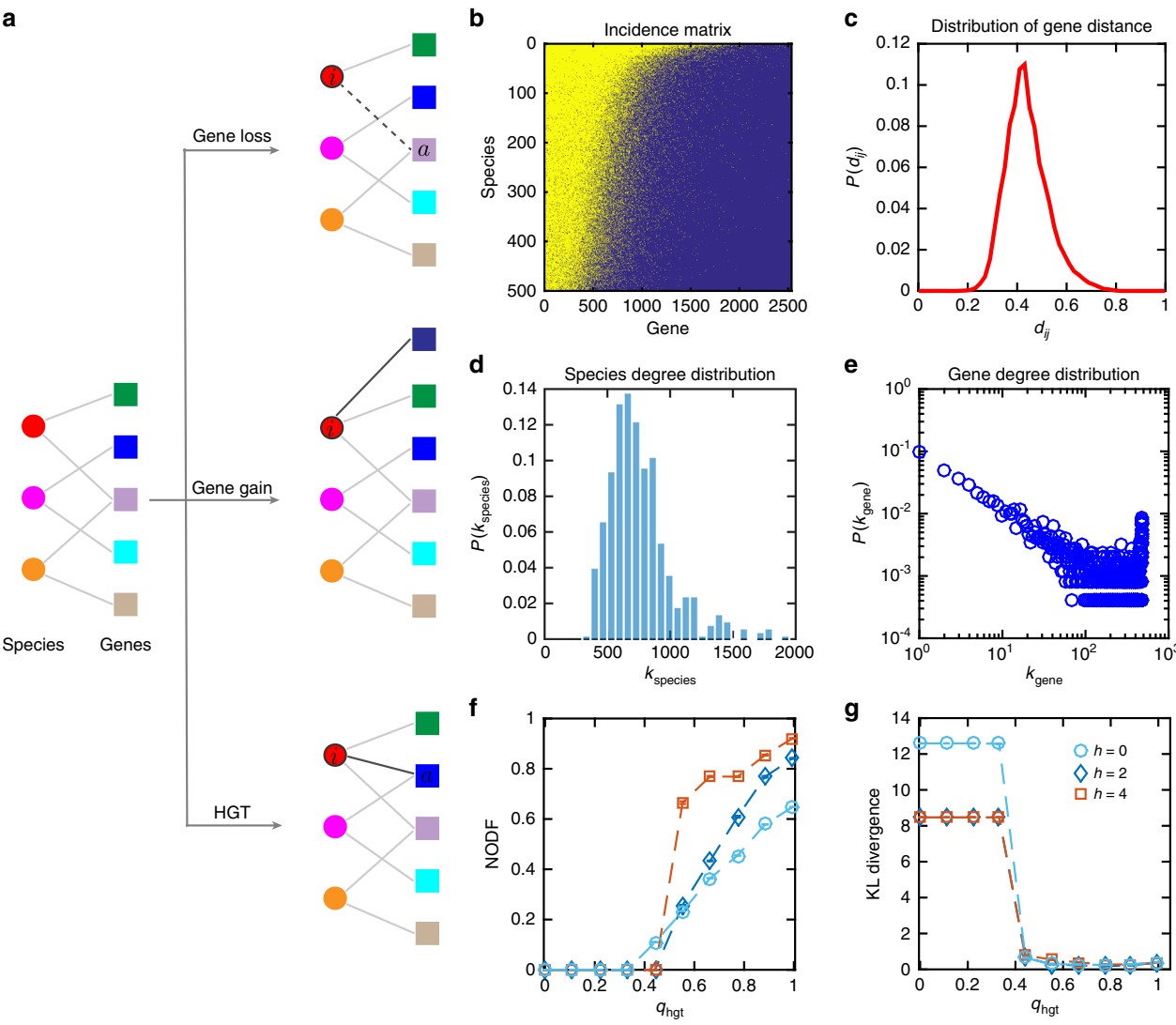

**Fig. 4 A simple genome evolution model can generate GCNs that capture key topological features of the real GCN. a** Schematic diagram of the genome evolution model. At each time step $t$, the genome of a species $i$ (shown in red) randomly chosen with probability proportional to $k_i^h$ will be updated based on one of the following three events: gene loss, gene gain, and horizontal gene transfer (HGT), with corresponding rates $q_{gl}$, $q_{gg}$, $q_{HGT}$, respectively. Note that the parameter $h \geq 0$ representing the selection pressure, and $h = 0$ corresponds to the case of neutral model. The three rates naturally satisfy $q_{gl} + q_{gg} + q_{HGT} = 1$. During HGT, a gene $a$ from a randomly chosen donor species is randomly selected and then transferred to the genome of species $i$. During gene loss, a gene $a$ in the genome of species $i$ is randomly selected and then removed. During gene gain, a new gene is added to the genome of species $i$. The initial GCN is a random bipartite graph that consists of 500 species and 200 genes with connection probability 0.8. The total number of evolution time steps is $5 \times 10^5$. **b–e** The incident matrix of the final GCN (with nestedness value NODF = 0.703), functional distance, species degree and gene degree distributions with $h = 2$, $q_{HGT} = 0.795$, $q_{gg} = 0.005$, and $q_{gl} = 0.2$ (See Supplementary Figs. 12, 13 for those topological features with other model parameters). **f** The nestedness (quantified by NODF) of the final GCN calculated as a function of HGT rate with different selection pressure $h = 0,2,4$. **g** The Kullback–Leibler (KL) divergence between the normalized gene degree distribution $P(\tilde{k}_{gene})$ of real GCN and that of the simulated GCNs calculated with different selection pressures and HGT rates as shown in **f**. Here the normalized gene degree $\tilde{k}_{gene} \equiv k_{gene}/k_{gene}^{max}$.

sequencing data and tools from network science. In future application of the GCN framework, it should be straight forward to apply our computational pipeline to ask how within-sample FR varies with changing environment. Such studies will require high-quality longitudinal data with changing environmental factors such as dietary alterations.

## Methods

**Genomic content network**. Consider a metacommunity of $N$ taxa and $M$ genes in total. Denote the taxonomic profile of a local community (e.g., the microbiome sample from a particular body site of subject v) as $\mathbf{p}^{(\nu)} = [p_1^{(\nu)}, \cdots, p_N^{(\nu)}]$, where $p_i^{(\nu)}$ is the relative abundance of the $i$th taxon and $\sum_{i=1}^N p_i^{(\nu)} = 1$. Denote the gene composition (or functional profile) of this local community as

$\mathbf{f}^{(\nu)} = \left[ f_1^{(\nu)}, \cdots, f_M^{(\nu)} \right]$, where $f_a^{(\nu)}$ is the relative abundance of the $a$th gene and $\sum_{a=1}^M f_a^{(\nu)} = 1$. The GCN can be represented by an $N \times M$ incidence matrix $\mathbf{G} = [G_{ia}]$, where $G \geq 0$ is the copy number of gene-$a$ in the genome of taxon-$i$. The GCN naturally connects the taxonomic profile and the functional profile as follow:

$$\mathbf{f}^{(\nu)} = c\mathbf{p}^{(\nu)} \cdot G \tag{5}$$

or equivalently

$$f_a^{(\nu)} = c\sum_{i=1}^N p_i^{(\nu)} G_{ia} \tag{6}$$

for $a = 1, \ldots, M$. Here $c = \left[ \sum_{a=1}^M \sum_{i=1}^N p_i^{(\nu)} G_{ia} \right]^{-1}$ is the normalization constant.

**Functional distances measures**. In the main text the functional distance $d_{ij}$ between taxon-$i$ and $j$ is calculated as the weighted Jaccard distance between the

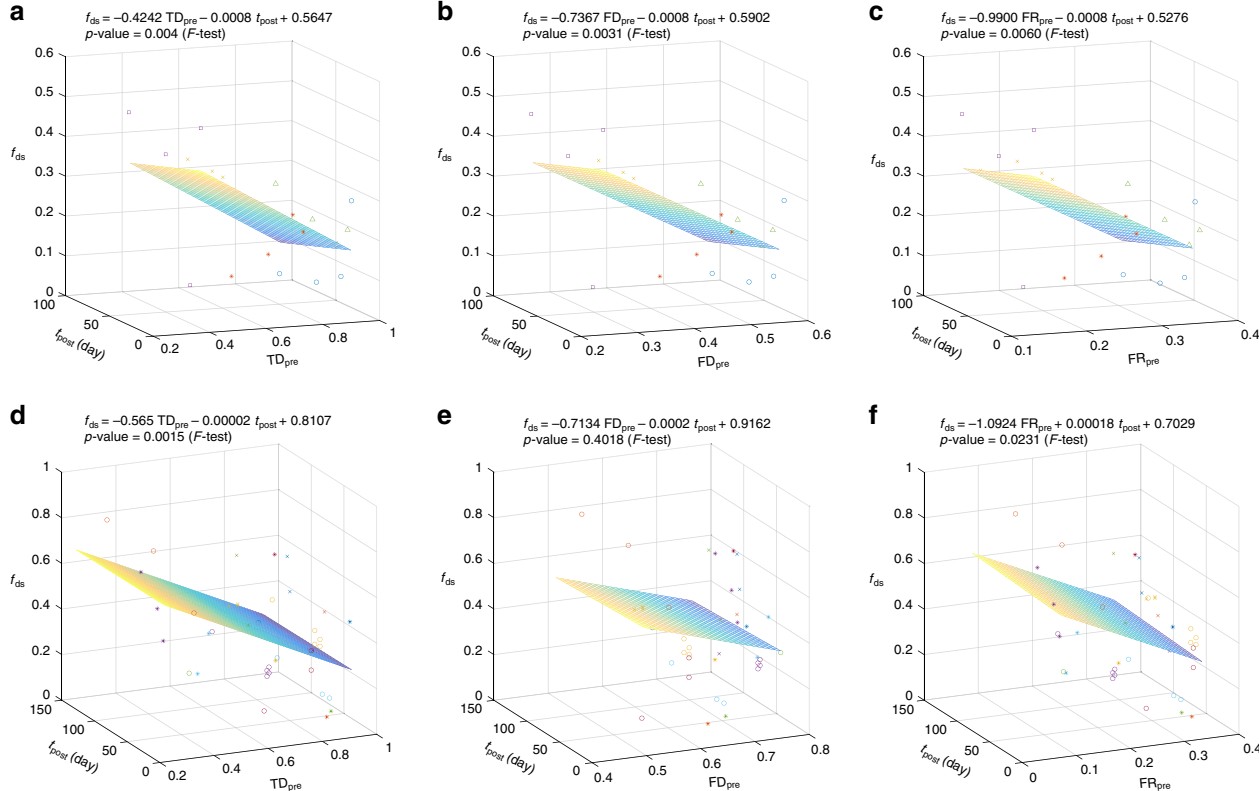

**Fig. 5 Functional redundancy of recipient's pre-FMT microbiota strongly affects the engraftment of donor microbiota.** Analysis of two published FMT studies: **a–c** Li et al., Science (2016), where each of the five patients with metabolic syndrome (represented by different symbols/colors) received a single FMT from one of three donors[44]; **d–f** Smillie et al., Cell Host & Microbe (2018), where each of the 19 patients with recurrent *C. difficile* infection (represented by different symbols/colors) were treated with FMT from one of four donors[45]. For each patient, we calculated: **a**, **d** the taxonomic diversity (TD) using the Gini-Simpson index; **b**, **e** the functional diversity (FD) using Rao's quadratic entropy; and **c**, **f** the functional redundancy (FR = TD-FD) of his/her pre-FMT gut microbiota, and the fraction of donor-specific strains at different time points post-FMT. We then performed multiple linear regression of the fraction of donor-specific strains as the response on TD (or FD, FR) of recipient's pre-FMT microbiota and the days post-FMT as the predictors. P values were calculated from F test.

genomes of the two taxa:

$$d_{ij} = 1 - \frac{\sum_a \min(G_{ia}, G_{ja})}{\sum_a \max(G_{ia}, G_{ja})}. \quad (7)$$

The reasons why we used the weighted Jaccard distance are twofold: (1) the genome of each taxon is represented by a vector of gene copy numbers, which are non-negative integers. Weighted Jaccard distance can naturally measure the distance between two vectors of non-negative integers. (2) The weighted Jaccard distance is normalized: $d_{ij} = 0$ indicates that taxon-$i$ and taxon-$j$ share exactly the same genome; $d_{ij} = 1$ means that they have totally different genomes. Other distance or dissimilarity measures that satisfy the above conditions can also be used, such as the correlation distance[58],

$$d_{ij}^{corr} = 1 - \frac{\sum_a G_{ia}G_{ja}}{\sqrt{\left(\sum_a G_{ia}^2\right)\left(\sum_a G_{ja}^2\right)}} \quad (8)$$

or the Sørensen dissimilarity[59],

$$d_{ij}^{Sørensen} = 1 - \frac{2\sum_a \min(G_{ia}, G_{ja})}{\sum_a G_{ia} + \sum_a G_{ja}}. \quad (9)$$

We have checked that our results do not change quantitatively by using different distance (or dissimilarity) measures (see Supplementary Fig. 1).

**Nestedness**. As a classical concept in ecology, nestedness characterizes the nested structure of ecological systems, such as the species-site network (describing the distribution of species across geographic locations), and the species-species interaction networks (e.g., host–parasite, plant–pollinator interactions)[32,60–65]. Roughly speaking, an ecological system is said to be nested if the items belonging to "smaller" elements (e.g., a small island containing few species, or a specialist species with few interactions) tend to be a subset of the items belonging to "larger" elements (e.g., a large island containing many species, or a generalist species with many interacting partners). Mathematically, those ecological systems can be

represented as bipartite graphs with two types of nodes, e.g., sites and species, hosts and parasites, plants and pollinators, etc. In this work, we focus on the GCN of microbial communities, which is also a bipartite graph with two types of nodes: species and genes.

Consider a general bipartite graph with N type-1 nodes and M type-2 nodes. The structure of this bipartite graph can be represented by its $N \times M$ binary incidence matrix $\mathbf{B} = (B_{ia})$, where $B_{ia} = 1$ if there is a link connecting the $i$th type-1 node and the $a$th type-2 node, and 0 otherwise. Mathematically, nestedness can be defined as a property of the incidence matrix $\mathbf{B}$. If there exists a permutation of rows and columns such that the set of links in row-i contains the links in row-($i$ + 1), and the set of links in column-$a$ contains those in column-($a$ + 1), then $\mathbf{B}$ is a perfectly nested binary matrix. For example, consider the mainland and a series of islands sorted according to their distances to the mainland. The mainland contains all the species, the first island has a subset of species in the mainland, the second island has a subset of species in the first island, etc.

**Numerical calculation of nestedness**. To quantify and visualize the nested structure of the incidence matrix, we can use the Nestedness Temperature Calculator (NTC)[31] based on the BINMATNESS algorithm[66], which also provides a nestedness measure. But NTC is time consuming for large incidence matrices. In this work we adopt the classical Nestedness metric based on Overlap and Decreasing Fill (NODF) to characterize the nested structure of a general bipartite graph[32]. Comparing with alternative nestedness measures, NODF reduces potential bias owing to network size and shape.

For a given bipartite graph (say, the genomic content or species-gene network) with binary incidence matrix $\mathbf{B}$, the (unweighted) degree of the $i$th species node is $k_i = \sum_{a=1}^M B_{ia}$, and the (unweighted) degree of the $a$th gene node is $k_a = \sum_{i=1}^N B_{ia}$. The number of common genes shared by the genomes of the $i$th and the $j$th species is given by $P_{ij} = \sum_{a=1}^M B_{ia}B_{ja}$. Similarly, the number of common species that both the $a$th and the $b$th genes appear in their genomes is given by $Q_{ij} = \sum_{i=1}^N B_{ia}B_{ja}$.

Define $\tilde{P}_{ij} = 0$ if $k_i = k_j$, and $\tilde{P}_{ij} = P_{ij}/\min(k_i, k_j)$ otherwise. Similarly, $\tilde{Q}_{ab} = 0$ if $k_a = k_b$, and $\tilde{Q}_{ab} = Q_{ab}/\min(k_a, k_b)$ otherwise. The NODF measure is defined as follows:

$$\text{NODF} = \frac{\sum_{i<j}^{N}\tilde{P}_{ij} + \sum_{a<b}^{M}\tilde{Q}_{ab}}{\frac{N(N-1)}{2} + \frac{M(M-1)}{2}}. \tag{10}$$

**Theoretical analysis of nestedness.** To theoretically analyze the nested structure of a given bipartite graph, one can construct a grand canonical ensemble for this bipartite graph under the constraint that, for the two types of nodes, the degree sequences in the ensemble match on average the empirical ones[67]. This theoretical approach has two big advantages. First, constraining the ensemble's mean degree sequence to be equivalent to the empirical one limits the possible effects of noisy data, hence possible missing (false negative) or overrated (false positive) links can be dealt with appropriately. Second, for this bipartite graph ensemble one can analytically derive the mean and standard deviation of the distribution of any network property (such as the classical NODF measure of nestedness) that can be analytically formulated in terms of the elements of the bipartite adjacency matrix **B**.

We applied this approach to the reference GCN shown in Fig. 2. We found that the expected nestedness of the grand canonical ensemble of randomized GCNs is 0.340581 (with standard deviation 0.001634), which is significantly lower than that of the real GCN (0.34712). One sample $z$ test yields $p$ value $= 6.2853 \times 10^{-5}$. This indicates that nestedness of the real GCN is an irreducible feature, which cannot be fully determined by the degree sequence of species and genes in the GCN.

**Microbiome data sets analyzed in this paper.** The microbiome data analyzed in this work are all from published studies. The original experiments and corresponding power analysis have been reported in previous publications. (1) HMP[13,39,40]. We analyzed the shotgun metagenomic sequencing data of the human microbiome samples from HMP. We focused on six body sites in five areas: the gut (one site: stool (549 samples)); the nasal cavity (one site: anterior nares (87 samples)); the oral cavity (two sites: buccal mucosa (368 samples) and tongue dorsum (418 samples)); the skin (one site: retroauricular crease (36 samples)); the vagina (one sites: posterior fornix (52 samples)). (2) Metagenomics of the Human Intestinal Tract (MetaHIT)[1,41]. We analyzed the shotgun metagenomic sequencing data of fecal samples from 177 healthy adults from MetaHIT. (3) FMT study of Li et al.[44] A cohort of five subjects (metabolic syndrome patients) received a single allogenic FMT from one of three lean donors unrelated to the recipients. Stool samples were collected from the donors (three samples) and five recipients before FMT (five samples) and after FMT at the 2nd, 14th, 42nd, and 84th days (20 samples). (4) FMT study of Smillie et al.[45] The cohort consist of 19 recurrent *C. difficile* patients. Feces from one of four donors were transplanted to each patient. Stool samples were collected from the donors (six samples) and the recipients before FMT (19 samples), and in follow-up visits ranging from 1 day to 4 months after FMT (40 samples). For FR calculation with reference genomes, samples with less than five strains with known genomes were excluded for analysis. For FR calculation without reference genomes (de novo method), no data were excluded (See Supplementary Sec. 2 for details).

**Reporting summary.** Further information on research design is available in the Nature Research Reporting Summary linked to this article.

## Data availability
The microbiome metagenomic data analyzed in this work are all from published studies: HMP[13,39,40] (Sequence data are available from the HMP DACC, http://hmpdacc.org); MetaHIT[1,41] (Sequence data are available from the European Nucleotide Archive under the accession code ERP002061); FMT study of Li et al.[44] (Sequence data are available from the European Nucleotide Archive under the accession code PRJEB12357); FMT study of Smillie et al.[45] (Sequence data are available from the European Nucleotide Archive under the accession code PRJEB23524). The HMP reference genomes are available from the Integrated Microbial Genomes and Microbiome (IMG/M) database (https://img.jgi.doe.gov/). The KEGG Orthologs and pathways are available at KEGG database (https://www.genome.jp/kegg/). The Clusters of Orthologous Groups of proteins are available at https://www.ncbi.nlm.nih.gov/research/cog-project/. The MicroBial Genome Database is available at http://mbgd.genome.ad.jp/. Example data analyzed in this work are available at https://github.com/liangtian85/FR. Source data are provided with this paper.

## Code availability
MATLAB Codes (version R2016b) used in this work are available at https://github.com/liangtian85/FR[68].

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

## Acknowledgements
We thank Yandong Xiao, Eugene Koonin, Patrick Chain, Uri Gophna for valuable discussions. Special thanks to Brigid Davis and Edwin K. Silverman for a careful reading of the manuscript. Y.-Y.L. acknowledges grants from National Institutes of Health (R01AI141529, R01HD093761, UH3OD023268, U19AI095219, and U01HL089856).

## Author contributions
Y.-Y.L. conceived and designed the project. L.T., X.-W.W., A.-K.W., Y.-H.F., and Y.-Y.L. did the analytical and numerical calculations. L.T. analyzed all the empirical data. X.-W.W. developed the genome evolution model. All authors interpreted the results. Y.-Y.L. wrote the manuscript. L.T., X.-W.W., A.-K.W., A.D., J.F., M.K.W., G.M.W., and S.T.W. edited the manuscript.

## Competing interests
The authors declare no competing interests.
