## [Peer Review File · Nature Communications]

REVIEWER COMMENTS

Reviewer #1 (Remarks to the Author):

The authors present a way to measure “functional redundancy” from microbiome meta-genomic data. They analyze the underlying gene-species networks and find that they show high nestedness and that the gene degree distribution is fat-tailed. These results are robust with respect to various calculation methods. The authors further reproduce these features in a simple genome evolution model with horizontal gene transfer. Finally, analyzing a small published dataset of several samples from fecal microbiota transplantation, they show that higher pre-FMT functional redundancy is correlated with lower fraction of donor strains in post-FMT samples.

My main problem with this paper is that it continuously confuses observations about microbiome composition with observations stemming solely from the species-gene network (which can be derived from genome databases without any need for microbiome data). Even though the authors motivate the study by being able to measure, understand and predict effects of functional redundancy in human microbiome samples, their analysis falls short of this important and intriguing goal. Instead of reflecting features of microbiome compositions, it seems that many of the results in this paper reflect merely the genome structure of bacteria and their relatedness. I will give two examples:

1. In Figure 2, the authors analyze the “reference” gene content network (GCN), yet this network is pretty much equivalent to analyzing known genomes and their relatedness in terms of genes or functions (namely, it is possible to get almost the same results just from the known whole-genome databases, even in absence of any microbiome samples). The observed nestedness of the GCN (Fig 2b) could potentially be explained by the genetic relatedness of bacterial species (without the microbiome). Indeed, apart from certain housekeeping gene functions which are shared by all taxa, there would be more gene functions which are shared among groups of species, e.g. gram positives vs. gram negatives or aerobes vs. anaerobes. Therefore, the taxonomic hierarchy alone (without any microbiome sample data) could likely explain much of the nestedness of these as well as individual sample profiles. To put in more mathematical/statistical language: by virtue of the methodology, any microbiome can only “pick and choose” a subset of the species-gene links that are predetermined by the genome data. Therefore, all the “features” of the network necessarily reflect features of this predetermined network.
2. The comparison of the features of the microbiome sample GCN to null models is artificial. As noted above, a GCN where new links between species and gene functions are formed cannot reliably be measured in a microbial sample with the methods used, so any model that does not take this into

consideration (Null-GCN-1 to 4) is not a relevant comparison (Fig 3a,b, S1, S4, S9). The Null-composition models are more relevant, but out of those, only Null-composition #3 preserves overall abundance of species across samples, which seems a much more relevant factor to control when trying to simulate other plausible assembly patterns, then the within-sample abundance profile (Null-composition #1 and #2). And the sample GCNs are either only slightly or non-significantly different in the features measured than this relevant null model (Fig 3c,d, S1, S4, S9) suggesting that the authors cannot strongly detect any microbiome assembly tendencies which would act to increase functional redundancy in the microbiome, compared to a reasonable random model.

Another important point:

In the interpretation of the FMT data, the small sample size of 5 patients is a drawback. Also, since the author's measure of FR is directly related to taxonomic diversity, which is easier to measure and commonly used as a predictor of microbiome stability, it is crucial to compare their result to a FMT success prediction purely based on taxonomic diversity.

Minor comments:

There are several places where the explanation is unclear or imprecise making understanding difficult:

The writing style is often strange for a normal scientific paper. For instance, the use of "one can..." to describe the methodology.

The usage of "selection pressure" in the genome evolution model is very confusing. The authors artificially enforce a tunable parameter controlling the extent to which longer genomes are more likely to gain and lose genes. This parameter is related to the rate

in which variations are formed (by mutation or horizontal gene transfer) not to the selection acting on these variations.

Various places, e.g. page 6 line 15, title of supp. section 3, refer to whole-genome sequencing when surely metagenomic shot-gun sequencing is meant based on the datasets described.

Reviewer #2 (Remarks to the Author):

This paper puts the human microbiome into an ecological context, this is very relevant and interesting. The genomic content network is an excellent way to look at functioning. I have several comments that are more to be better discussed than explicitly corrected. generally, I like the paper a lot.

p 3 | 14-15: When you introduce the GCN, first define what is that and then discuss what it is good for.

p 3 | 21: this is not a "metacommunity", this is a "species pool", ecologically speaking. The metacommunity is the population of the local communities (all humans or all samples).

p 5: in the beginning, you mention the individual-level variability of the human microbiome. Then, it is unclear here how to construct a universal GCN.

p 5 | 20: Why did you use the unweighted distribution? This results in a bias towards "rare contents".

p 8 | 26-27: The evolutionary model is a great idea but reproducing two, quite general topological features is not very convincing, I think. Also, I would suggest to look at the time series as well, not only at the endpoint. This could provide very useful information about the origins.

Following the ecological literature, key microbes could be defined as either central or unique. The former is of key importance because of being involved in multiple interactions, the latter is of key importance because of being non-replaceable (non-redundant).

For more details on this, see:

<https://royalsocietypublishing.org/doi/full/10.1098/rsbl.2011.1167>

I think this measure is more relevant here than the nestedness measure used in the manuscript. You may consider checking it out.

When discussing the GCN, you may mention that it is possible because the genotype * phenotype (G*P) mapping is relatively simple for prokaryotes. In case of higher organisms, the gene content and the function are not simply related.

Can the degree of a bacterium species/strain/OTU be related to its evolutionary age? Older organisms might be more connected (accumulating genes)? Or less (lost genes)? Any pattern in your evolution model?

Some empirical and comparative work could be helpful in order to better understand the "minimal genomic content", and also to better understand the least redundant species/strains/OTUs who can establish "bacterial monocultures" in the body, as far as possible.

You mention it but it should be better elaborated: diversity and redundancy are mostly considered here from a metabolic viewpoint but the bacterial niche is of higher dimension. The "habitat" and maybe also some "founder" or "sequence" effect can also be of crucial importance. This is why there can be a huge diversity also between body parts, see:

<https://www.nature.com/articles/srep15920>

<https://journals.sagepub.com/doi/abs/10.1177/0022034509346811>

For the bottom-up explanation, you mention competition. Why not cooperation and even community-level selection? These have been richly discussed in the literature.

From an ecological point of view, it makes sense that higher FR is more sensitive against invaders (transplants) but these systems should be also more healthy (resistant against other microbes), so they probably need less transplants. It would be nice to identify systems that are unhealthy but resistant against the needed transplants.

Functional redundancy and diversity should be understood also in terms of environmental variability. This might be related to moderate selection pressure. Too strong selection destroys the needed diversity, while too weak cannot put together well-functioning systems, see:

Turnbaugh, P. J. et al. The human microbiome project: exploring the microbial part of ourselves in a changing world. *Nature* 449, 804–810 (2007).

Beyond a set of bacteria (and a set of their genes), somehow their interaction networks could be considered too. If there is a metabolic chain, one missing gene is not only one out of many on the list but a missing link in the chain. This should be better discussed. See:

Berry, D. & Widder, S. Deciphering microbial interactions and detecting keystone species with co-occurrence networks. *Front.*

Microbiol. 5, 219 (2014).

Faust, K. et al. Microbial co-occurrence relationships in the human microbiome. PLoS Comput. Biol. 8(7), e1002606 (2012).

My final question if there is any way to map the properties of the GCN and particular diseases.

Reviewer #3 (Remarks to the Author):

The authors investigated potential mechanisms driving functional redundancy in the microbiome using previously published data from HMP, MetaHIT and a metabolic disease cohort. This is an extremely novel and interesting concept that was well investigated using appropriate statics and tools from various disciplines, especially network science. The authors considered potentially limiting factors including choice of index, body site, etc and reached similar conclusions indicating the functional redundancy is favored. This article provides important insights into factors associated with microbiota resilience that will be important to consider as microbiota therapeutics are developed.

Responses to Reviewer #1

The authors present a way to measure “functional redundancy” from microbiome meta-genomic data. They analyze the underlying gene-species networks and find that they show high nestedness and that the gene degree distribution is fat-tailed. These results are robust with respect to various calculation methods. The authors further reproduce these features in a simple genome evolution model with horizontal gene transfer. Finally, analyzing a small published dataset of several samples from fecal microbiota transplantation, they show that higher pre-FMT functional redundancy is correlated with lower fraction of donor strains in post-FMT samples.

We thank Reviewer #1 for her/his comprehensive review of our work. We next address each of her/his concerns in order.

My main problem with this paper is that it continuously confuses observations about microbiome composition with observations stemming solely from the species-gene network (which can be derived from genome databases without any need for microbiome data). Even though the authors motivate the study by being able to measure, understand and predict effects of functional redundancy in human microbiome samples, their analysis falls short of this important and intriguing goal. Instead of reflecting features of microbiome compositions, it seems that many of the results in this paper reflect merely the genome structure of bacteria and their relatedness. I will give two examples:

We thank Reviewer #1 for this critical comment.

Here, we would like to emphasize that the within-sample FR of any microbiome sample is **jointly** determined by the structure of the underlying gene content network (GCN) and the sample’s microbial composition. This can be seen from the definition of within-sample FR. In this work, we defined the within-sample FR, i.e., the alpha functional redundancy of a microbiome sample (or local community) to be the part of its alpha taxonomic diversity (TD_α) that cannot be explained by its alpha functional diversity (FD_α); i.e.,

$$FR_\alpha \equiv TD_\alpha - FD_\alpha.$$

If we choose TD_α to be the Gini-Simpson index, i.e.,

$$TD_\alpha \equiv 1 - \sum_{i=1}^N p_i^2 = \sum_{i=1}^N \sum_{j \neq i}^N p_i p_j$$

representing the probability that two randomly chosen members of the local community belong to two different taxa; and choose FD_α to be the Rao’s quadratic entropy, i.e.,

$$FD_\alpha \equiv \sum_{i=1}^N \sum_{j \neq i}^N d_{ij} p_i p_j$$

characterizing the mean functional distance between any two randomly chosen members in the local community; then we have

$$FR_\alpha = \sum_{i=1}^N \sum_{j \neq i}^N (1 - d_{ij}) p_i p_j,$$

naturally representing the functional similarity (or overlap) of two randomly chosen members in the local community. Here $d_{ij} = d_{ji} \in [0, 1]$ denotes the functional distance between taxon- i and taxon- j , which can be calculated as the weighted Jaccard distance between the genomes of the two taxa. By definition, $d_{ii} = 0$ for $i = 1, \dots, N$.

From the above equation of FR_α , we can see clearly that FR_α of any microbiome sample is **jointly** determined by two factors: (1) the functional distances d_{ij} 's among taxa present in the sample, which are predetermined by the structure of the sample-specific GCN; and (2) the microbial composition or taxonomic profile $\mathbf{p} = [p_1, \dots, p_N]$ of this microbiome sample. Note that this is true even if we consider more complicated definitions of taxonomic (and functional) diversity measures, e.g., a parametric class of taxonomic (or functional) diversity measures based on Hill numbers (see SI, Sec.2. Eqs.S26-S27).

In the revised manuscript, we have emphasized this point as follows (see **main text, page 5, lines 2-5**):

“From Eq.[4] we can see clearly that FR_α of any microbiome sample is jointly determined by two factors: (1) the functional distances d_{ij} 's among taxa present in the sample, which are predetermined by the structure of the GCN; and (2) the microbial composition or taxonomic profile $\mathbf{p} = [p_1, \dots, p_N]$ of this microbiome sample.”

1. In Figure 2, the authors analyze the “reference” gene content network (GCN), yet this network is pretty much equivalent to analyzing known genomes and their relatedness in terms of genes or functions (namely, it is possible to get almost the same results just from the known whole-genome databases, even in absence of any microbiome samples).

We thank Reviewer #1 for this critical comment. We fully agree with her/him that the reference GCN can be studied without any microbiome samples. We apologize for not presenting the motivations of studying the reference GCN more clearly in the previous version. There are actually several motivations of studying the reference GCN alone, i.e., in the absence of any microbiome samples.

- *First*, as we explained above, the within-sample FR of any microbiome sample is **jointly** determined by the **structure of the sample-specific GCN** and the **sample's microbial composition**. Hence, it is very natural to ask which of the two factors plays a more important role in determining the within-sample FR of human microbiome samples. To disentangle the effects of the two factors, we have to study them separately.
- *Second*, any sample-specific GCN is just a subgraph of the reference GCN. Instead of studying the structure of all the sample-specific GCNs (which will be quite challenging to demonstrate), it is very natural to first study the structure of the reference GCN.

- *Third*, the relatedness of known microbial genomes (especially those microbes colonized in and on human body) has never been studied purely from the network science perspective. In fact, the term of **reference GCN** has never been coined before. And this paper is the first time for us to visualize this network and report its highly nested structure.

Taken together, studying the structure of the reference GCN alone (without microbiome samples) is both novel and meaningful.

The observed nestedness of the GCN (Fig 2b) could potentially be explained by the genetic relatedness of bacterial species (without the microbiome). Indeed, apart from certain housekeeping gene functions which are shared by all taxa, there would be more gene functions which are shared among groups of species, e.g. gram positives vs. gram negatives or aerobes vs. anaerobes. Therefore, the taxonomic hierarchy alone (without any microbiome sample data) could likely explain much of the nestedness of these as well as individual sample profiles.

We fully agree with Reviewer #1 that the observed nested structure of the reference GCN can be conceptually/qualitatively explained by the genetic relatedness of bacterial species and taxonomic hierarchy. But in this work we want to offer a more rigorous/quantitative understanding of this intriguing structural feature.

- *First*, we emphasize that the highly nested structure of the reference GCN cannot be simply accounted for by housekeeping genes. As shown in main text Fig.2b, the incidence matrix G_{ia} of the GCN still displays a highly nested structure even in the absence of housekeeping genes (the leftmost yellow columns).
- *Second*, if we randomize the GCN but preserve the gene degree distribution, the randomized GCNs have much lower nestedness than that of the real GCN (SI, Fig.S3).
- *Third*, we adopted tools from statistical physics to calculate the expected nestedness value and its standard deviation for an ensemble of randomized GCNs in which the expected species and gene degree distributions match those of the real GCN. We found that the expected nestedness of randomized GCNs is significantly lower than that of the real GCN (one sample t-test yields $p_{\text{value}} = 6.2853 \times 10^{-5}$, see SI Sec.1.3.2 for details).

Taken together, the highly nested structure of the reference GCN is non-trivial. We hope that our presented results improved the understanding of this intriguing structure.

To put in more mathematical/statistical language: by virtue of the methodology, any microbiome can only “pick and choose” a subset of the species-gene links that are predetermined by the genome data. Therefore, all the “features” of the network necessarily reflect features of this predetermined network.

We fully agree with Reviewer #1. As we explained above, any sample-specific GCN is just a subgraph of the reference GCN. Hence, features of sample-specific GCNs will reflect features of the reference GCN, and vice versa. Since demonstrating the reference GCN is much easier and more convenient than demonstrating hundreds of sample-specific GCNs, we chose to demonstrate the structural features of the reference GCN.

In the SI (Fig.S5), we did demonstrate the structural features of several body site-specific GCNs. Here, a body site-specific GCN is equivalent to the merging of all the sample-specific GCNs for samples from the same body site. Any body site-specific GCN is still a subgraph of the reference GCN, but it can be considered as an intermediate level in the hierarchy of sample-specific GCNs ---- body site-specific GCNs ---- the reference GCN. We constructed several body site-specific GCNs from two metagenomic datasets: MetaHIT and HMP, finding that those body site-specific GCNs display similar structural features as the reference GCN, which agrees well with our expectation.

2. The comparison of the features of the microbiome sample GCN to null models is artificial. As noted above, a GCN where new links between species and gene functions are formed cannot reliably be measured in a microbial sample with the methods used, so any model that does not take this into consideration (Null-GCN-1 to 4) is not a relevant comparison (Fig 3a,b, S1, S4, S9).

We thank Reviewer #1 for this critical comment. We apologize for not clearly explaining our motivation of introducing those null GCN models.

- *First*, as shown in the definition $FR_{\alpha} = \sum_{i=1}^N \sum_{j \neq i}^N (1 - d_{ij}) p_i p_j$, the within-sample FR of any microbiome sample is **jointly** determined by d_{ij} 's (which are determined by the structure of the underlying GCN) and the sample's microbial composition. To disentangle the effects of the two factors on the within-sample FR, we have to study them separately. That's why we introduced two different types of null models: null GCN models and null composition models.
- *Second*, to study the impact of the GCN on FR_{α} of a microbiome sample, we need to fix the microbial composition $\mathbf{p} = [p_1, \dots, p_N]$ of the sample, and then randomize the GCN. In principle, for each sample we can randomize its sample-specific GCN. But mathematically, this is equivalent to randomizing the body site-specific GCN (or the reference GCN), because any sample-specific GCN is just a subgraph of the body site-specific GCN (or the reference GCN). In our calculations (Fig. 3a-b in the main text), for each body site, we just randomized the body site-specific GCN, rather than those individual sample-specific GCNs.
- *Third*, since GCN is a bipartite graph connecting species and their genes, there are at least four different randomization schemes corresponding to four different null models: Complete randomization (Null-GCN-1); Species-degree preserving randomization (Null-GCN-2); gene-degree preserving randomization (Null-GCN-3); Species- and gene-degree preserving randomization (Null-GCN-4). **Those null models are of course all artificial, but the whole**

point of comparing FR_α calculated from the real GCN with that calculated from those artificial GCNs is to figure out which structural features of the real GCN are truly important to determine the value of FR_α . Indeed, we made two interesting discoveries:

- All the four different null models yield lower nFR_α (Fig.3a-b, colored boxes) than those calculated from real GCNs (Fig.3a-b, black boxes) for all the body sites examined in this work. Analyzing the network properties of those null models (SI, Figs.S7-S8), we found that those randomized GCNs all display lower nestedness and higher $\langle d_{ij} \rangle$ than those of the real GCNs. **Thus, the highly-nested structure and low $\langle d_{ij} \rangle$ of the real GCNs contribute to the high nFR_α values observed in human microbiome samples.**
- Moreover, for the first two null models (Null-GCN-1 and Null-GCN-2, where both the highly-nested structure and high gene degree heterogeneity of the real GCN are destroyed), nFR_α is much lower than those of the other two null models (Null-GCN-3 and Null-GCN-4, where the highly-nested structure is destroyed, but the high gene degree heterogeneity is kept). **This suggests that the high gene degree heterogeneity also contributes to the high nFR_α values of human microbiome samples.**

Taken together, randomizing the real GCN while fixing microbial composition is an efficient way to reveal the structural features of the real GCN that are truly important to determine FR_α . Our results indicate that the real GCN exhibits at least three different structural features (highly nested structure, low $\langle d_{ij} \rangle$, and heterogeneous gene degree distribution) that jointly contribute to the high nFR_α values of human microbiome samples.

The Null-composition models are more relevant, but out of those, only Null-composition #3 preserves overall abundance of species across samples, which seems a much more relevant factor to control when trying to simulate other plausible assembly patterns, then the within-sample abundance profile (Null-composition #1 and #2). And the sample GCNs are either only slightly or non-significantly different in the features measured than this relevant null model (Fig 3c,d, S1, S4, S9) suggesting that the authors cannot strongly detect any microbiome assembly tendencies which would act to increase functional redundancy in the microbiome, compared to a reasonable random model.

We thank Reviewer #1 for this critical comment. We apologize for not clearly explaining our results calculated from the three null composition models.

To test if the microbe assemblages or their abundances play an important role in determining nFR_α , we randomized the compositions of human microbiome samples using three different randomization schemes corresponding to three different null composition models:

- Randomized microbial assemblage generated by randomly choosing the same number of species from the species pool but keeping the species abundance profile unchanged (Null-composition-1);

- Randomized microbial abundance profiles through random permutation of non-zero abundance for each sample across different species (Null-composition-2);
- Randomized microbial abundance profiles through random permutation of non-zero abundance for each species across different samples (Null-composition-3).

Before we explain our results, we would like to make two remarks.

Remark 1: Just like the null GCN models, all the null composition models are artificial. We fully agree with Reviewer #1 that only Null-composition-3 preserves the overall abundance of species across samples. In other words, Null-composition-3 is more “realistic” than Null-composition-1 and Null-composition-2.

Remark 2: For Null-composition-2 and Null-composition-3, the sample-specific GCN is fixed. For Null-composition-1, the sample-specific GCN is actually different from that of the real microbiome sample (because species in the sample are randomly replaced by species from the species pool). But we argue that the key structure features (e.g., highly nested structure, low $\langle d_{ij} \rangle$, and heterogeneous gene degree distribution) will be preserved, even after the species replacement. In other words, Null-composition-1 still preserves the key structure features of the sample-specific GCN, which also reflect the features of the reference GCN.

The point of introducing the three different null composition models is that they allow us to reveal which aspect (microbe assemblages or their abundances) plays a more important role in determining the within-sample FR. Indeed, we found that Null-composition-1 always yields much lower nFR_α (Fig.3c-d, colored boxes) than that of the real composition (Fig.3c-d, black box). Interestingly, the other two models (Null-composition-2 and Null-composition-3) did not always yield much lower nFR_α than that of the original composition.

Taken together, randomizing the microbial compositions while fixing key structure features of the GCN allows us to reveal microbiome assembly tendencies that would act to increase FR in the human microbiome samples. In particular, our findings suggest that:

- **the species present in each microbiome sample are not assembled randomly, but follow certain assembly rules that maximize its within-sample FR.** Revealing the detailed assembly rules of human microbiome requires a very careful analysis of large-scale longitudinal datasets, which is beyond the scope of the current research. We hypothesize that those assembly rules might be body site-specific.
- **the assemblage of microbes plays a more important role than their abundances in determining the high FR of the human microbiome.** We hypothesized that the specific environment (e.g., the host nutrient and immune state) from particular microbiome samples were obtained will tend to select for sets of functions among most or all inhabitants, at any abundance.

Another important point:

In the interpretation of the FMT data, the small sample size of 5 patients is a drawback.

We thank Reviewer #1 for this critical comment. We fully agree with Reviewer #1 that the small sample size of 5 patients in the FMT study [R1] is a drawback. To directly address this issue, in the revised manuscript we also analyzed another published FMT study [R2], where 19 patients with recurrent *C. difficile* infection (rCDI) were treated with FMT from one of four donors. We found that results from this FMT study of a much larger sample size (which provides increased statistical power) are qualitatively consistent with that from the FMT study of a small sample size we analyzed before (**see Fig.R1**).

Also, since the author's measure of FR is directly related to taxonomic diversity, which is easier to measure and commonly used as a predictor of microbiome stability, it is crucial to compare their result to a FMT success prediction purely based on taxonomic diversity.

We thank Reviewer #1 for this very insightful and constructive comment.

In Fig.R1, we analyzed data from two published FMT studies [R1,R2]. For each individual in the two FMT studies, we plotted the fraction of donor-specific strains (denoted as f_{ds}) as a function of (1) the time post-FMT (denoted as t_{post}); and (2) the TD (FD or FR) of the pre-FMT gut microbiota, denoted as TD_{pre} (FD_{pre} or FR_{pre} , respectively). We then performed multiple linear regression that uses t_{post} and TD_{pre} (FD_{pre} or FR_{pre}) as the explanatory variables to predict the outcome of the response variable f_{ds} .

We found that the association between f_{ds} and FR_{pre} is much stronger than that between f_{ds} and TD_{pre} (or FD_{pre}). Moreover, F-test revealed significant negative association between f_{ds} and FR_{pre} (or TD_{pre} , but not FD_{pre}) in both studies. Note that the significance of the association between f_{ds} and FD_{pre} is not conclusive from the two studies: The FMT study of metabolic syndrome with a small sample size ($n=5$) suggests significant association; while the FMT study of rCDI with a large sample size ($n=19$) suggests nonsignificant association. We think the latter is more convincing because of its much larger sample size. Of course, this issue deserves more dedicated large-scale clinical studies, which are beyond the scope of the current work.

Taken together, our results are consistent with the fact pointed by Reviewer #1 that TD is commonly used as a predictor of microbiome stability. Moreover, since $TD = FD+FR$ by definition, our results indicate that the microbiome stability could be largely explained by FR, rather than FD. Indeed, the association between f_{ds} and FR_{pre} is much stronger than that between f_{ds} and TD_{pre} (or FD_{pre}). So, FR can serve as a stability or resilience indicator of microbiome.

Minor comments:

There are several places where the explanation is unclear or imprecise making understanding difficult:

We thank Reviewer #1 for pointing this out. We have carefully revised the manuscript to

improve its readability.

The writing style is often strange for a normal scientific paper. For instance, the use of “one can...” to describe the methodology.

We thank Reviewer #1 for pointing this out. We have revised the manuscript accordingly.

The usage of “selection pressure” in the genome evolution model is very confusing. The authors artificially enforce a tunable parameter controlling the extent to which longer genomes are more likely to gain and lose genes. This parameter is related to the rate in which variations are formed (by mutation or horizontal gene transfer) not to the selection acting on these variations.

We thank Reviewer #1 for this critical comment.

In our genome evolution model, at each time step t , we randomly select a species i with probability $p_i = k_i^h / \sum_{j=1}^N k_j^h$, and then update its genome based on one of the following three events: gene loss, gene gain, and horizontal gene transfer (HGT), with corresponding rates q_{gl} , q_{gg} , q_{HGT} , respectively. Here, k_i is the degree (i.e., the genome size) of species i ($i = 1, \dots, N$), and $h \geq 0$ is a tuning parameter.

We think h represents the “selection pressure” simply because for any $h > 0$, species with larger genome sizes are more likely to be chosen to update their genomes, while the case $h = 0$ corresponds to the neutral model where all the species have equal probability to be chosen to update their genomes. In other words, positive h just means that selection pressure favors changes in larger genomes.

In principle, we can also introduce “selection pressure” acting on the three events separately: gene loss, gene gain, and HGT. But we feel this will result in a more complicated model with more parameters. So, in a sense, we are offering a minimal model to reproduce the key structural features of the real GCN. In the revised manuscript, we clearly mention this point.

Various places, e.g. page 6 line 15, title of supp. section 3, refer to whole-genome sequencing when surely metagenomic shot-gun sequencing is meant based on the datasets described.

We thank Reviewer #1 for pointing this out. Throughout the revised manuscript, we have replaced “whole-genome sequencing (WGS)” by “shotgun metagenomic sequencing (SMS)”.

Finally, we thank Reviewer #1 again for her/his very insightful and constructive comments. We hope our responses above have addressed those very legitimate issues/concerns in a satisfactory manner.

Responses to Reviewer #2

This paper puts the human microbiome into an ecological context, this is very relevant and interesting. The genomic content network is an excellent way to look at functioning. I have several comments that are more to be better discussed than explicitly corrected. generally, I like the paper a lot.

We thank Reviewer #2 for reviewing our manuscript and her/his very positive assessment on the relevance and general interest of our work. We next address each of her/his concerns in order.

p 3 | 14-15: When you introduce the GCN, first define what is that and then discuss what it is good for.

We thank Reviewer #2 for this constructive comment. In the revised manuscript (**main text, page 3, lines 19-23**), we have added the definition and properties of the GCN as follows:

“In particular, we constructed the gene content network (GCN) of the human microbiome, which is a bipartite graph connecting microbes to the genes in their genomes. The GCN provides a full description of the functional overlap of different microbes in microbial communities, which enables us to quantify the within-sample FR for any given human microbiome sample for the first time.”

p 3 | 21: this is not a "metacommunity", this is a "species pool", ecologically speaking. The metacommunity is the population of the local communities (all humans or all samples).

We thank Reviewer #2 for pointing this out. We have revised that sentence accordingly.

p 5: in the beginning, you mention the individual-level variability of the human microbiome. Then, it is unclear here how to construct a universal GCN.

We thank Reviewer #2 for this critical comment. We apologize for not making this clear in the previous version. Indeed, the taxonomic profiles of human microbiome samples are highly personalized. But we can construct a reference GCN based on the pool of human-associated microbes identified from the Human Microbiome Project (HMP). In particular, we constructed a reference GCN using the Integrated Microbial Genomes & Microbiomes (IMG/M) database, focusing on the HMP-generated metagenome datasets. This IMG/M-HMP database used includes in total 1,555 strains (genomes) and 7,210 KEGG Orthologs (KOs).

In the revised manuscript (**main text, page 6, lines 1-3**), we added the following sentence.

“Although the taxonomic profiles of human microbiome samples are highly personalized, we can construct a reference GCN based on the pool of human-associated microbes to quantitatively study the GCN underlying the human microbiome.”

p 5 | 20: Why did you use the unweighted distribution? This results in a bias towards "rare contents".

We thank Reviewer #2 for pointing this out. Here, the unweighted degree of a species is just the number of distinct genes in its genome, while the unweighted degree of a gene is the number of species whose genomes contain this gene. The unweighted degree distributions were just used to demonstrate the structure features of the GCN. When calculating the within-sample FR, we always use the weighted incidence matrix $G = (G_{ia})$ of the GCN, where a non-negative integer G_{ia} indicates the copy number of gene- a in the genome of taxon- i . This way, there will be no bias towards "rare contents".

p 8 | 26-27: The evolutionary model is a great idea but reproducing two, quite general topological features is not very convincing, I think. Also, I would suggest to look at the time series as well, not only at the endpoint. This could provide very useful information about the origins.

We thank Reviewer #2 for this comment. As shown in **SI Fig.S11**, our model can actually reproduce four topological features as observed in the real GCN: highly nested structure, unimodal functional distance d_{ij} distribution, power-law degree distribution of genes, Poisson degree distribution of species. The emergence and evolution of those topological features can also be found in the “time series” shown in **SI Fig.S11**.

Following the ecological literature, key microbes could be defined as either central or unique. The former is of key importance because of being involved in multiple interactions, the latter is of key importance because of being non-replaceable (non-redundant).

For more details on this, see:

<https://royalsocietypublishing.org/doi/full/10.1098/rsbl.2011.1167>

I think this measure is more relevant here than the nestedness measure used in the manuscript. You may consider checking it out.

We thank Review #2 for this very insightful comment.

- *First of all*, we would like to point out that nestedness is a measure for the whole GCN, rather than any specific microbe. Hence, nestedness offers a global network perspective that cannot be provided by any single node.
- *Second*, we fully agree with Reviewer #2 that the importance of a species can be quantified by measuring the centrality of its position in an ecological network, where nodes represent different species and edges represent direct ecological interactions between different species (e.g., parasitism, commensalism, mutualism, amensalism, or competition). But the

GCN defined in this work is fundamentally different from any ecological network in literature. In the GCN, nodes represent species and genes, links represent the presence (and copy number) of a gene in the genome of a particular species. It is very challenging, if not impossible, to infer inter-species interactions based on the GCN, because the genome similarity or difference of different species can result in different types of ecological interactions, e.g., competition and mutualism.

- *Third*, inspired by Reviewer #2's comment, we can rank the importance of species based on their **functional substitutability**, defined as $s_i = \sum_{j \neq i} (1 - d_{ij}) / (N - 1) \in [0, 1]$, where $d_{ij} = d_{ji} \in [0, 1]$ denotes the functional distance between species-*i* and species-*j*, which can be calculated as the weighted Jaccard distance between the genomes of the two species. By definition, $d_{ii} = 0$ for $i = 1, \dots, N$. Note that $s_i = 0$ if species-*i* is completely non-substitutable, i.e., the functional distance $d_{ij} = 1$ between species-*i* and any other species; while $s_i = 1$ if species-*i* is completely substitutable, i.e., the functional distance $d_{ij} = 0$ between any two species. The relationship between the sample-specific functional substitutability s_i and relative abundance p_i across different samples is shown in **Fig.R2, row-1**. For each sample, we can calculate the Pearson correlation coefficient r between s_i and p_i . The distribution of r across different samples is shown in **Fig.R2, row-2**. We find that for some body sites (e.g., gut and tongue dorsum), most of the samples display slightly positive correlation between s_i and p_i , implying that highly abundant species tend to have high functional substitutability. For other body sites (e.g., buccal mucosa and posterior fornix), s_i and p_i do not display strong correlation. We also calculated the body site-specific functional substitutability as a function of the average relative abundance for strain-*i* across all samples with $p_i > 0$. The result (as shown in **Fig.R2, row-3**) is consistent with what we found in row-2.

In the revised manuscript (**main text, page 12, lines 9-19**), we have cited the paper mentioned by Reviewer #2 and added the following discussion:

“In an ecological network, the importance of a species can be quantified by measuring the centrality⁵² of its position in the network, where nodes represent different species and edges represent direct ecological interactions between different species (e.g., parasitism, commensalism, mutualism, amensalism, or competition)^{53,54}. We emphasize that the GCN defined here is fundamentally different from the ecological networks in literature. In the GCN, nodes represent species and genes, and links represent the presence (and copy number) of a gene in the genome of a particular species. It is very challenging, if not impossible, to infer inter-species interactions based on the GCN because there is clear relationship between the genome similarity of different species and their ecological interactions. Similarly, it might be very challenging to infer species abundance correlation⁵⁵ or co-occurrence^{56,57} simply based on the GCN.”

When discussing the GCN, you may mention that it is possible because the genotype * phenotype (G*P) mapping is relatively simple for prokaryotes. In case of higher organisms, the gene content and the function are not simply related.

We thank Review #2 for this excellent suggestion. We fully agree with Reviewer #2 that the genotype-phenotype mapping is relatively simple for prokaryotes, which enables us to relate gene content and functional capacity. In the revised manuscript (**main text, page 5, lines 26-29**), we added the following sentence:

“Note that the genotype-phenotype mapping is relatively simple for prokaryotes, which enables us to relate their gene content and functional capacity. For higher organisms, their gene content and function capacity are not simply related, which means that the GCN framework cannot be naively applied to study the FR of communities of high organism.”

Can the degree of a bacterium species/strain/OTU be related to its evolutionary age? Older organisms might be more connected (accumulating genes)? Or less (lost genes)? Any pattern in your evolution model?

We thank Reviewer #2 for this very insightful comment.

In our genome evolution model, for the sake of simplicity, we fix the total number of species during the evolution, indicating that all the species have the same age. In this model, at each time step t , we will randomly select a species i with probability $p_i = k_i^h / \sum_{j=1}^N k_j^h$, and then update its genome based on one of the following three events: gene loss, gene gain, and horizontal gene transfer (HGT), with corresponding rates q_{gl} , q_{gg} , q_{HGT} , respectively. Here, k_i is the degree (i.e., the genome size) of species i ($i = 1, \dots, N$), and $h \geq 0$ is a tuning parameter. We think h represents the “selection pressure” because for any $h > 0$, species with larger genome sizes are more likely to be chosen to update their genomes, while the case $h = 0$ corresponds to the neutral model, where all the species have equal probability to be chosen to update their genomes. In other words, positive h just means that selection pressure favors changes in larger genomes.

We can certainly extend this model to incorporate the **speciation of a new species**. To achieve that, we still start from an initially random constructed bipartite graph with certain number of species and genes. At each time step $t > 0$, we will randomly select a species i with probability $p_i = k_i^h / \sum_{j=1}^N k_j^h$ to either copy its genome for the **speciation** of a new species or update its genome based on one of the three events: **gene loss**, **gene gain**, and **HGT** (see **Fig.R3A**). Those four events occur with the corresponding rates q_s , q_{gl} , q_{gg} , q_{HGT} , respectively. During speciation, all of genes of an existing or old species will be inherited by a new species. The “age” of each species can then be quantified by the time steps elapsed since its speciation. This way those species created at earlier are “older” than those created later.

For this extended model, the relationship between species degree and age was shown in **Fig.R3B-G**. With $q_s = 0.001$, $q_{gl} = 0.199$, $q_{gg} = 0.005$, $q_{HGT} = 0.795$, we found that for the neutral model ($h = 0$), the species degree is almost independent of species age (**Fig.R3B**, Pearson correlation $r = 0.02$, p -value= 0.57, paired t-test). For weak selection pressure ($h = 2$), the species degree negatively correlates with species age (Fig.R3C, $r = -0.25$, p -value= 7.41×10^{-16}). For strong selection pressure ($h = 4$), the species degree displays very strong negative correlation with species age (**Fig.R3D**, $r = -0.81$, p -value= 7.28×10^{-233}). These results are quite robust against model parameters. For a different set of $q_s, q_{gl}, q_{gg}, q_{HGT}$ values, we observed the same phenomenon (**Fig.R3E,F,G**).

The phenomenon that “older” species tend to be less connected (or equivalently, “newer” species tend to be more connected) can be understood as follows. In the extended model, a new species is created by copying all the genes of an existing old species. In the presence of selection pressure ($h > 0$), species with higher degree (more genes) are more likely to be selected for speciation to create a new species. In other words, the new species also tend to have higher degree. This trend is robust against different parameters, because it is determined by the effect selection pressure in the model.

Some empirical and comparative work could be helpful in order to better understand the "minimal genomic content", and also to better understand the least redundant species/strains/OTUs who can establish "bacterial monocultures" in the body, as far as possible.

We thank Reviewer #2 for this very constructive comment.

Regarding the concept “minimal genomic content”, it arose from the observations that many genes do not appear to be essential for the survival of an organism, which can therefore be removed from its genome. The minimal set of genes (which is closely related to the notion of housekeeping genes) required for the basic metabolism and replication of the organism can be obtained by experimental and computational analysis of the biochemical pathways needed to carry out basic metabolism and reproduction. This is certainly beyond the scope of the current work. But we would like to point out that **our GCN framework does allow us to “approximately” identify those housekeeping genes.** First of all, those genes that appear as the leftmost yellow columns in the GCN’s incidence matrix (Fig.2b) appear in almost every species’ genome. This core or common set of genes is required for diverse microbes to perform basic cellular functions and/or survive in the host body site they inhabit. Hence, they are housekeeping genes. Second, as shown in Fig.2e, the gene degree distribution is highly heterogeneous and displays a fat tail, indicating that most genes are specialized and only exist in the genomes of very few species, and a few housekeeping genes appear in almost every species’ genome to maintain basic cellular functions.

Regarding the concept of the “least redundant species”, as mentioned above, we can rank the importance of species based on their functional substitutability, defined as $s_i = \sum_{j \neq i} (1 - d_{ij}) /$

$(N - 1)$. The species with the lowest s_i corresponds to the least redundant species, which is most non-substitutable. Note that $s_i = 0$ means species- i is completely non-substitutable, i.e., the functional distance $d_{ij} = 1$ between species- i and any other species. As shown in **Fig.R2**, for some body sites (e.g., gut and tongue dorsum), most of the samples display slightly positive correlation between s_i and p_i , implying that highly abundant species tend to have high functional substitutability. For other body sites (e.g., buccal mucosa and posterior fornix), s_i and p_i do not display strong correlation.

You mention it but it should be better elaborated: diversity and redundancy are mostly considered here from a metabolic viewpoint but the bacterial niche is of higher dimension. The "habitat" and maybe also some "founder" or "sequence" effect can also be of crucial importance. This is why there can be a huge diversity also between body parts, see:
<https://www.nature.com/articles/srep15920>
<https://journals.sagepub.com/doi/abs/10.1177/0022034509346811>

We thank Reviewer #2 very much for pointing out those important references. We have cited them appropriately in the revised manuscript.

In this work, we focused on the quantification and interpretation of the FR of human microbiome samples. The phenomenon of **population-level FR** (i.e., the gene composition or functional capacity of the human microbiome is highly conserved across individuals, while the taxonomic composition varies tremendously across individuals) has been observed for different body sites. What we achieved in this work is to quantify the **within-sample FR** for human microbiome sample. Our computational pipeline can also be generalized to analyze any microbiome samples.

By introducing different null composition models, we were able to reveal which aspect (microbe assemblages or their abundances) plays a more important role in determining the within-sample FR. In particular, we found that (1) the species present in each microbiome sample are not assembled randomly, but follow certain assembly rules that maximize its within-sample FR; (2) the assemblage of microbes plays a more important role than their abundances in determining the high FR of the human microbiome.

We hypothesized that the specific environment (e.g., the host nutrient and immune state) from which particular microbiome samples were obtained will tend to select for sets of functions among most or all inhabitants, at any abundance.

We fully agree with Reviewer #2 that the colonization history (or the priority effect) plays a very important role shaping the human microbiome and could be used to explain the huge diversity of microbial communities across different body sites. But the lack of high-quality data on the colonization history of any human body site renders quantitative studies very challenging. We feel this is beyond the scope of the current paper and will leave it as a future work.

For the bottom-up explanation, you mention competition. Why not cooperation and even community-level selection? These have been richly discussed in the literature.

We thank Reviewer #2 for this very insightful comment.

We fully agree with Reviewer #2 that cooperation and community-level selection of microorganisms have been richly discussed in the literature. Here, the “bottom-up” selection pressure is purely discussed from the perspective of individual microorganisms, rather than the whole community. From the perspective of individual microorganism, those species with similar genomes (functional capacities) will tend to occupy the same ecological niche and hence more likely compete with each other. Such competitions between members of the microbiota will exert “bottom-up” selection pressure that results in specialized genomes with functionally distinct suites of genes.

From an ecological point of view, it makes sense that higher FR is more sensitive against invaders (transplants) but these systems should be also more healthy (resistant against other microbes), so they probably need less transplants. It would be nice to identify systems that are unhealthy but resistant against the needed transplants.

We thank Reviewer #2 very much for this very constructive comment.

Indeed, it would be very nice to identify diseased microbiome that are resistant against transplants. As we know, fecal microbiota transplantation (FMT) is increasingly being explored as a potential treatment to optimize microbiota composition and functionality. But recurrent *Clostridioides difficile* infection (rCDI) is so far the only disease that has the most robust clinical evidence supporting the use of FMT [R3]. Numerous case reports and cohort studies have described the use of FMT in patients with inflammatory bowel disease (IBD), but the efficacy is not comparable to the case of rCDI [R4]. To understand the efficacy difference from the ecological point of view, we calculated the normalized functional redundancy (nFR) of the gut microbiome samples from a CDI cohort [R5] and an IBD cohort [R6]. The metagenomic data of both cohorts were downloaded from the curated metagenomic data package [R7].

The results are presented in **Fig.R4**. We found that CDI patients exhibited a much lower nFR median value than that of their controls. The relatively low nFR of CDI patients is expected to reduce the resilience of their gut microbiota against external perturbation, potentially facilitating the efficacy of FMT in restoring a healthy gut microbiota. Unlike those of CDI patients, the stool samples from IBD patients do not differ significantly from their respective controls in their nFR values. Thus, in contrast to CDI patients, the gut microbiomes of IBD patients might have been driven to relatively stable diseased states, which are resilient to compositional transitions and hence less likely to be susceptible to FMT that “engineers” the microbial community as a whole.

Taken together, we think the gut microbiota of IBD patients could represent such an unhealthy system that is resistant against the needed transplants. We hypothesize that many other chronic

diseases (that have been associated with disrupted microbiota but lack robust clinical evidence supporting the use of FMT) could also represent such an unhealthy but resistant system.

Functional redundancy and diversity should be understood also in terms of environmental variability. This might be related to moderate selection pressure. Too strong selection destroys the needed diversity, while too weak cannot put together well-functioning systems, see: Turnbaugh, P. J. et al. The human microbiome project: exploring the microbial part of ourselves in a changing world. *Nature* 449, 804–810 (2007).

We thank Reviewer #2 for this very insightful comment. We fully agree with her/him that functional redundancy and diversity should be understood in terms of environmental variability. That's actually the exact next step of our research. In the current work, our primary goal is to establish the GCN framework, validate the computational pipeline, and explain the high FR observed in the human microbiome, using cross-sectional shotgun metagenomic sequencing (SMS) data and tools from network science. It would be a very natural question to ask how FR varies with changing environment. But this would require high-quality longitudinal SMS data or SMS data with different environmental factors. Both types of data are not easily accessible. Hence, we will leave this as a future work.

In the revised manuscript (**main text, page 12, lines 20-26**), we added the following discussion:

“In the current work, our primary goal was to establish the GCN framework, validate the computation pipeline of within-sample FR calculation, and explain the high FR observed in the human microbiome, using cross-sectional shotgun metagenomic sequencing data and tools from network science. In future application of the GCN framework, it should be straight forward to apply our computational pipeline to ask how within-sample FR varies with changing environment. Such studies will require high-quality longitudinal data with changing environmental factors such as dietary alterations.”

Beyond a set of bacteria (and a set of their genes), somehow their interaction networks could be considered too. If there is a metabolic chain, one missing gene is not only one out of many on the list but a missing link in the chain. This should be better discussed. See:

Berry, D. & Widder, S. Deciphering microbial interactions and detecting keystone species with co-occurrence networks. *Front. Microbiol.* 5, 219 (2014).

Faust, K. et al. Microbial co-occurrence relationships in the human microbiome. *PLoS Comput. Biol.* 8(7), e1002606 (2012).

We thank Reviewer #2 very much for pointing out those important references. We have cited them appropriately in the revised manuscript.

As we explained above, the GCN defined in this work is fundamentally different from any ecological network in literature. In the GCN, nodes represent species and genes, links represent

the presence (and copy number) of a gene in the genome of a particular species. It is very challenging, if not impossible, to infer inter-species interactions based on the GCN, because the genome similarity or difference of different species can result in different types of ecological interactions, e.g., competition and mutualism. In the revised manuscript (**main text, page 12, lines 9-19**), we have explicitly discussed the difference between the ecological network and the GCN.

Inspired by Reviewer #2's comment, we explored the relationship between the functional distance d_{ij} and abundance correlation (denoted as c_{ij}) between species- i and species- j across samples in the Human Microbiome Project analyzed in this work. We found that for all the different body sites studied here, there is no strong correlation between d_{ij} and c_{ij} (**Fig.R5**). This result implies that it is very challenging, if not impossible, to infer species abundance correlation simply based on the GCN. We think this could be due to the fact that the genome/functional difference of different species can result in different types of ecological interactions, e.g., competition and mutualism.

My final question if there is any way to map the properties of the GCN and particular diseases.

We thank Reviewer #2 for this very insightful comment.

We constructed disease-specific GCNs from gut microbiome samples in a CDI and an IBD cohort, respectively. The results are presented in **Fig.R4b,c,e,f**. Unfortunately, we didn't find any particular disease-specific GCN properties.

We also compared the normalized functional redundancy (nFR) of cases and controls in each cohort (**Fig.R4a,d**). As mentioned earlier, the gut microbiomes of CDI patients exhibited a much lower nFR median value than that calculated for their controls. By contrast, the gut microbiomes of IBD patients do not differ significantly from their respective controls in their nFR values.

Taken together, nFR is a more sensitive measure than GCN properties in distinguishing cases and controls for certain disease (such as CDI). We think this is because nFR takes into account both the GCN structure and the disrupted microbial compositions associated with different diseases.

Finally, we thank Reviewer #2 again for her/his very insightful and constructive comments. We hope our responses above have addressed those very legitimate issues/concerns in a satisfactory manner.

Responses to Reviewer #3

The authors investigated potential mechanisms driving functional redundancy in the microbiome using previously published data from HMP, MetaHIT and a metabolic disease cohort. This is an extremely novel and interesting concept that was well investigated using appropriate statistics and tools from various disciplines, especially network science. The authors considered potentially limiting factors including choice of index, body site, etc and reached similar conclusions indicating the functional redundancy is favored. This article provides important insights into factors associated with microbiota resilience that will be important to consider as microbiota therapeutics are developed.

We thank Reviewer #3 for reviewing our paper and her/his very positive assessment on the importance and novelty of our work. We share her/his belief that this work will offer important insights to understand the resilience of human microbiota under perturbation such as microbiota therapeutics.

Figure R1: Functional redundancy of recipient’s pre-FMT microbiota strongly affects the engraftment of donor microbiota. Analysis of two published FMT studies: **(a-c)** Li et al., *Science* (2016), where each of the 5 patients with metabolic syndrome (represented by different symbols/colors) received a single FMT from one of three donors; **(d-f)** Smillie et al., *Cell Host & Microbe* (2018), where each of the 19 patients with recurrent *C. difficile* infection (represented by different symbols/colors) were treated with FMT from one of four donors. For each patient, we calculated: **(a,d)** the taxonomic diversity (TD) using the Gini-Simpson index; **(b,e)** the functional diversity (FD) using Rao’s quadratic entropy; and **(c,f)** the functional redundancy (FR=TD-FD) of his/her pre-FMT gut microbiota, and the fraction of donor-specific strains at different time points post-FMT. We then performed multiple linear regression of the fraction of donor-specific strains as the response on TD (or FD, FR) of recipient’s pre-FMT microbiota and the days post-FMT as the predictors. P-values were calculated from F-test.

Figure R2: The relationship between the functional substitutability and taxonomic abundance. (Row-1) Sample-specific functional substitutability (s_i) vs. relative abundance (p_i) for strains in 5 randomly chosen samples from MetaHIT (for gut) and HMP (for six different body sites). For each sample, we can calculate the Pearson correlation coefficient r between s_i and p_i . **(Row-2)** Histograms of Pearson correlation coefficients r 's for different samples. Red/blue bars indicate positive/negative r , respectively. **(Row-3)** Body site-specific functional substitutability as a function of the average relative abundance for strains across all samples with $p_i > 0$. The solid lines represent the trend calculated using the Robust Locally-Weighted Regression (LOWESS) method.

Figure R3: The extended genome evolution model that explicitly considers speciation. **A**, the schematic diagram of this model. At each time step $t > 0$, we will randomly select a species i with probability $p_i = k_i^h / \sum_{j=1}^N k_j^h$ to either copy its genome for the speciation of a new species or update its genome based on one of the three events: gene loss, gene gain, and HGT. Those four events occur with the corresponding rates $q_s, q_{gl}, q_{gg}, q_{HGT}$, respectively. **B-G**: The relation between species degree and its “age” (the time steps elapsed after it was first created). **B-D**: $q_s = 0.001, q_{gl} = 0.199, q_{gg} = 0.005, q_{HGT} = 0.795$. **E-G**: $q_s = 0.001, q_{gl} = 0.039, q_{gg} = 0.01, q_{HGT} = 0.95$. **E**, $h = 0$. **F**, $h = 2$. **G**, $h = 4$. For each panel, r represents the Pearson correlation and p is the p-value calculated from paired t-test.

Figure R4: The normalized functional redundancy and structural properties of the genomic content networks (GCNs) for CDI (top) and IBD (bottom). The metagenomic data of both disease cohorts were downloaded from <https://waldronlab.io/curatedMetagenomicData>. (a,d), The normalized functional redundancy of control group and case group. The Mann–Whitney U test is performed to calculate the p values. (b1,c1,e1,f1), The incidence matrix of the GCN shown at the strain-KO level, where the presence (or absence) of a link between a strain and a KO is colored in yellow (or blue), respectively. We organized this matrix using the Nestedness Temperature Calculator to emphasize its nested structure. The nestedness of this network is calculated based on the NODF measure. (b2,c2,e2,f2), The distribution of functional distances between different strains. (b3,c3,e3,f3), The unweighted species degree distribution. (b4,c4,e4,f4), The unweighted KO degree distribution.

Figure R5: Functional distance vs. abundance correlation of samples from the Human Microbiome Project (for six different body sites). The species abundance correlations were calculated by SparCC [R8] using the SpiecEasi R package [R9]. The statistical test of the correlation between functional distance (d_{ij}) and abundance correlation (c_{ij}) was done by performing the Mantel test [R10]. Pearson correlation coefficient r and the p-value were shown in the top of each panel.

References

- [R1] Li, S. S., Zhu, A., Benes, V., Costea, P. I., Hercog, R., Hildebrand, F., Huerta-Cepas, J., Nieuwdorp, M., Salojärvi, J., Voigt, A. Y., Zeller, G., Sunagawa, S., De Vos, W. M. & Bork, P., Durable coexistence of donor and recipient strains after fecal microbiota transplantation. *Science* 352, 586-589 (2016).
- [R2] Smillie, C. S., Sauk, J., Gevers, D., Friedman, J., Sung, J., Youngster, I., Hohmann, E. L., Staley, C., Khoruts, A., Sadowsky, M. J., Allegretti, J. R., Smith, M. B., Xavier, R. J. & Alm, E. J., Strain Tracking Reveals the Determinants of Bacterial Engraftment in the Human Gut Following Fecal Microbiota Transplantation. *Cell Host and Microbe* 23, 229-240.e225 (2018).
- [R3] Kassam, Z., Lee, C.H., Yuan, Y. & Hunt, R.H. Fecal microbiota transplantation for *Clostridium difficile* infection: systematic review and meta-analysis. *The American journal of gastroenterology* 108, 500-508 (2013).
- [R4] Paramsothy, S. et al. Faecal Microbiota Transplantation for Inflammatory Bowel Disease: A Systematic Review and Meta-analysis. *J Crohns Colitis* 11, 1180–1199 (2017).
- [R5] Vincent, C., Miller, M. A., Edens, T. J., Mehrotra, S., Dewar, K. & Manges, A. R. Bloom and bust: intestinal microbiota dynamics in response to hospital exposures and *Clostridium difficile* colonization or infection. *Microbiome* 4, 12 (2016).
- [R6] Nielsen, H. B. et al., Identification and assembly of genomes and genetic elements in complex metagenomic samples without using reference genomes. *Nat Biotechnol* 32, 822-828 (2014).
- [R7] Pasolli, E., Schiffer, L., Manghi, P., Renson, A., Obenchain, V., Truong, D. T., Beghini, F., Malik, F., Ramos, M., Dowd, J. B., Huttenhower, C., Morgan, M., Segata, N. & Waldron, L. Accessible, curated metagenomic data through ExperimentHub. *Nature Methods* 14, 1023-1024 (2017).
- [R8] Friedman, J. & Alm, E. J. Inferring correlation networks from genomic survey data. *PLoS Comput Biol* 8, e1002687, (2012).
- [R9] <https://github.com/zdk123/SpiecEasi>
- [R10] <http://scikit-bio.org/docs/0.5.1/generated/generated/skbio.stats.distance.mantel.html>

REVIEWERS' COMMENTS

Reviewer #1 (Remarks to the Author):

I thank the authors for their careful and serious review of my earlier comments. The revised manuscript is more clear in several aspects. Indeed, it also now better acknowledges the dependency of the key results jointly on both the underlying gene content network (GCN) and the sample's microbial composition. Yet, this leaves me confused as to the relative contributions of the genome-based network and the microbiome composition which are still not resolved. I am suspicious/worried that most of the contribution comes from the gene content of the genomes and the contribution from the microbiome compositions is small. The manuscript inherently mixes and fails to clearly resolve these two very different types of observations (gene content of species, and microbiome composition). Overall, I feel that much of the results of this paper pertain to gene-content relatedness among microbial species rather than to species composition of the microbiome and to the question of how the species composition itself affects microbiome functional redundancy.

Reviewer #2 (Remarks to the Author):

You carefully and correctly answered all of my questions. The manuscript has been improved a lot. Thank you.

Responses to Reviewer #1

I thank the authors for their careful and serious review of my earlier comments. The revised manuscript is more clear in several aspects. Indeed, it also now better acknowledges the dependency for the key results jointly on both the underlying gene content network (GCN) and the sample's microbial composition.

We thank Reviewer #1 very much for reviewing our paper again. We are glad to know that s/he appreciated our responses to her/his previous comments.

Yet, this leaves me confused as to the relative contributions of the genome-based network and the microbiome composition which are still not resolved. I am suspicious/worried that most of the contribution comes from the gene content of the genomes and the contribution from the microbiome compositions is small. The manuscript inherently mixes and fails to clearly resolve these two very different types of observations (gene content of species, and microbiome composition). Overall, I feel that much of the results of this paper pertain to gene-content relatedness among microbial species rather than to species composition of the microbiome and to the question of how the species composition itself affects microbiome functional redundancy.

We thank Reviewer #1 for this comment. As we pointed out in the previous response letter and the revised manuscript, by definition, the functional redundancy (FR_α) of any microbiome sample is jointly determined by two factors: (1) the functional distances d_{ij} 's among taxa present in the sample, which are predetermined by the structure of the genomic content network (GCN); and (2) the microbial composition or taxonomic profile $p = [p_1, \dots, p_N]$ of this microbiome sample. Yet, this doesn't mean mathematically one can separate the FR_α of any microbiome sample into two independent and additive terms: one is purely contributed by GCN, and the other is purely contributed by the microbial composition. To study which of the two factors plays a more important role in determining the FR_α of microbiome samples, we have to "disentangle" the impacts of the two factors on FR_α in a more sophisticated way. In particular, to study the impact of GCN on the FR_α of a microbiome sample, we can fix its microbial composition and then randomize the GCN in different ways (rendering different null GCN models). Similarly, to study the impact of microbial composition on the FR_α of a microbiome sample, we can fix the GCN, and then randomize the microbial composition in different ways (rendering different null composition models). This is exactly what we did in this work.

We apologize for not making this point clear in the previous version. In the revised version (Results section), we added the following subsection (see **main text, page 7**) to explicitly state this point.

Disentangle impacts of GCN and microbial composition on FR. As mentioned above, FR_α of any microbiome sample is jointly determined by two factors: (1) the functional distances d_{ij} 's among taxa present in the sample that are predetermined by the structure of the GCN; and (2) the microbial composition $\mathbf{p} = [p_1, \dots, p_N]$ of this sample. Yet, this doesn't mean mathematically one can separate the FR_α of any microbiome sample into two independent and additive terms: one is purely contributed by GCN, and the other is purely contributed by the microbial composition. Indeed, as shown in Eq.[4] (or

Eqs.[S20-S21]), there is always a term in FR_α that involves the multiplication of d_{ij} and $p_i p_j$ (or their respective functions). This term cannot be separated into two independent and additive expressions of d_{ij} and $p_i p_j$, respectively. To study which of the two factors plays a more important role in determining the FR_α of microbiome samples, we have to “disentangle” the impacts of the two factors on FR_α in a more sophisticated way. To achieve that, in the following two subsections, we introduced two different types of null models: null GCN models and null composition models.

Responses to Reviewer #2

You carefully and correctly answered all of my questions. The manuscript has been improved a lot. Thank you.

We thank Reviewer #2 very much for reviewing our paper. We are very pleased to know that s/he is now happy with the revised version.